# OOD Link Prediction Generalization Capabilities of Message-Passing GNNs in Larger Test Graphs

**Yangze Zhou**
Department of Statistics
Purdue University
West Lafayette, IN 47903
zhou950@purdue.edu

**Gitta Kutyniok**
Department of Mathematics
Ludwig-Maximilians-Universitat München
Munich, Germany
kutyniok@math.lmu.de

**Bruno Ribeiro**
Department of Computer Science
Purdue University
West Lafayette, IN 47903
ribeiro@cs.purdue.edu

## Abstract

This work provides the first theoretical study on the ability of graph Message Passing Neural Networks (gMPNNs) —such as Graph Neural Networks (GNNs)— to perform inductive out-of-distribution (OOD) link prediction tasks, where deployment (test) graph sizes are larger than training graphs. We first prove non-asymptotic bounds showing that link predictors based on permutation-equivariant (structural) node embeddings obtained by gMPNNs can converge to a random guess as test graphs get larger. We then propose a theoretically-sound gMPNN that outputs structural pairwise (2-node) embeddings and prove non-asymptotic bounds showing that, as test graphs grow, these embeddings converge to embeddings of a continuous function that retains its ability to predict links OOD. Empirical results on random graphs show agreement with our theoretical results.

## 1 Introduction

Link prediction is the task of predicting whether two nodes likely have a missing link [1, 12, 30, 37, 66]. Link prediction tasks arise in many settings, ranging from predicting edges on bipartite graphs between users and products or content in recommender systems [6, 11, 31, 32, 39, 62], to knowledge graph reconstruction [4, 14, 20, 54, 66, 67], to predicting protein-protein interactions [57].

In recent years, there has been growing interest in applying neural network models to inductive link prediction tasks. Inductive link prediction considers methods trained on a graph $G^{\text{tr}}$ and deployed at test time on another graph $G^{\text{te}}$. It also encompasses the task of training the method on a smaller induced subgraph $G^{\text{tr}}$ of a larger graph $G^{\text{te}}$, then deploying it on the entire graph. In particular, our work focuses on graph message-passing Neural Networks (gMPNNs) [21, 60] or, more precisely, the widely used Graph Neural Network (GNN) framework [8, 9, 13, 22, 24, 28, 61, 64, 69].

Our work asks the following questions: *Are link prediction methods able to cope with the task of inductive out-of-distribution (OOD) link prediction, where (unseen) test graphs are significantly larger than training graphs?* How can these OOD link prediction tasks be theoretically defined? Can we obtain non-asymptotic bounds on the generalization capabilities of these methods?

The majority of today's link prediction methods are based on a similar principle. Consider an attributed graph $G = (V, E)$, with node set $V = \{1, ..., N\}$, edge set $E \subseteq V \times V$, and node features

---

36th Conference on Neural Information Processing Systems (NeurIPS 2022).

$\boldsymbol{F} \in \mathbb{R}^{N \times F_0}$, $F_0 \geq 1$. Then, given a pair of nodes $i, j \in V$, after $T \geq 1$ iterations over $G$, these methods produce associated node embeddings (representation vectors) $\Theta_i^\bullet, \Theta_j^\bullet \in \mathbb{R}^{F_T}$, $F_T \geq 1$, which are then used in a link function $\eta^\bullet : \mathbb{R}^{F_T} \times \mathbb{R}^{F_T} \to [0, 1]$ such that $\eta^\bullet(\Theta_i^\bullet, \Theta_j^\bullet)$ predicts the probability that $i$ and $j$ have a missing link in $G$. In our notation we will denote all node embeddings and associated functions with the superscript "$\bullet$". Henceforth we denote gMPNNs that output structural node embeddings as gMPNNs$^\bullet$.

*Node embeddings.* The first part of our work considers a subset of these methods, where the output node embeddings are permutation equivariant (a.k.a. *structural node embeddings* [65]). Informally, a sequence of node embeddings $\Theta^\bullet \in \mathbb{R}^{N \times F_T}$ given by an embedding method is permutation-equivariant if for any arbitrary graph $G$ and any permutation $\pi \in \mathbb{S}_N$ of the node indices, where $\mathbb{S}_N$ is the symmetric group, the resulting isomorphic graph $G' = (\pi \circ V, \pi \circ E, \pi \circ \boldsymbol{F})$ gets permuted node embeddings $\Theta^{\bullet\prime} = \pi \circ \Theta^\bullet$, where $\pi \circ M$ defines the action of $\pi$ on $M$ (we will provide a formal definition in Section 2). We leave the study of OOD link prediction with *positional node embeddings* (a.k.a. permutation-sensitive node embeddings [65]) to future work.

The application of GNNs to link prediction tasks is made difficult by the fact that, by construction, permutation-equivariant GNNs give the same embeddings $\Theta_i^\bullet, \Theta_j^\bullet$ to any isomorphic nodes $i, j$ in $G$, as noted by You et al. [82] and Srinivasan and Ribeiro [65]. Isomorphic nodes are nodes that are structurally indistinguishable in $G$ (even when considering node features) except by their (assumed arbitrary) node indices $i, j \in V$. That is, if a graph has isomorphic pairs, permutation-equivariant GNN link prediction can fail. The recent link prediction literature has significantly relied on isomorphic nodes for theoretical results (e.g., Zhang et al. [86, Theorem 2] uses isomorphic nodes to prove that, uniformly, graphs are likely to have many isomorphic nodes and hence are not amenable to accurate link prediction). However, isomorphic nodes are rare in both real-world graphs (see Figure 3 in the Appendix) and in large random graphs (Proposition 1).

An important open question is whether equivariant GNN would be able to predict links in asymmetric graphs. That is, the concerns of [65, 82] may not be of practical importance. Our work also answers this question: We see that for in-distribution link prediction tasks (where graph test sizes are the same as training sizes), permutation-equivariant GNNs are able to predict links by tapping into the graph asymmetries. However, we show theoretically and empirically that tapping into asymmetries can fail OOD even when it works in-distribution.

*Pairwise embeddings.* Taking a different route, Srinivasan and Ribeiro [65] provides an existence proof that the link prediction task between $i$ and $j$ can always be performed by a pairwise embedding $\Theta_{ij}^{\bullet\bullet}(G)$, i.e., for any pair of nodes $i, j$ in a graph $G$, there exists a pairwise embedding $\Theta_{ij}^{\bullet\bullet}(G)$ and a link function $\eta^{\bullet\bullet} : \mathbb{R}^{F_T} \to [0, 1]$ such that $\eta^{\bullet\bullet}(\Theta_{ij}^{\bullet\bullet})$ approximates the probability that $i$ and $j$ have a hidden link. In our notation we will denote all pairwise (joint 2-node) embeddings and associated functions with the superscript "$\bullet\bullet$". Unfortunately, as the test graph grows, we were unable to prove existing pairwise embedding methods [50, 72, 84, 86, 87] are able to perform OOD link prediction tasks. Hence, we propose a novel family of gMPNNs for pairwise embeddings, denoted gMPNNs$^{\bullet\bullet}$ henceforth. The second part of our work considers the OOD generalization capability of these gMPNNs$^{\bullet\bullet}$.

**Contributions.** In this work we study inductive OOD link prediction tasks for larger test graphs using permutation-equivariant node and pairwise embeddings, $\Theta^\bullet$ and $\Theta^{\bullet\bullet}$, respectively. Our work makes the following contributions:

1. We provide a theoretical framework defining OOD inductive link prediction tasks, where test graphs are significantly larger than training graphs.

2. We show that structural node embeddings from message-passing GNNs can fail in OOD link prediction tasks if the test graph (from the same graph family) is significantly larger than the training graph. Our work fills *an important gap in the literature*, where Bevilacqua et al. [7] studied the OOD capabilities of GNNs for *graph classification* using random graph models. Our work studies the OOD capabilities of GNNs for *inductive link prediction* in a similar setting.

3. We propose a new family of structural pairwise embeddings, denoted gMPNNs$^{\bullet\bullet}$, that can provably perform the above OOD task.

4. We provide non-asymptotic bounds on the convergence of pairwise gMPNNs embeddings. Extensive empirical experiments using stochastic block models (SBMs [63]) validate our theoretical

results. Our work focuses on providing a theoretical understanding of the challenges of OOD link prediction tasks rather than propose real-world link prediction tasks and compare baselines. However, we believe that our work lays the theoretical foundation (and challenges) for future application-focused works.

## 2 Preliminaries

Given an attributed graph $G = (V, E)$, with node set $V = \{1, ..., N\}$, edge set $E \subseteq V \times V$, adjacency matrix $\boldsymbol{A} \in \{0, 1\}^{N \times N}$, where $\boldsymbol{A}_{ij} = \mathbb{1}_{\{(i,j) \in E\}}$, and node features $\boldsymbol{F} \in \mathbb{R}^{N \times F_0}$, $F_0 > 0$. Let $\boldsymbol{P}_\pi \in \mathcal{B}_N$ be a permutation matrix associated with permutation $\pi \in \mathbb{S}_N$ (where $\mathbb{S}_N$ is the symmetric group), where $\mathcal{B}_N$ denotes the Birkhoff polytope of $N \times N$ doubly-stochastic matrices. Doubly-sctochastic matrices are non-negative square matrices whose rows and columns sum to one. The matrix $\boldsymbol{P}_\pi$ defines the action of permutation $\pi$ on these matrices, e.g., $\pi \circ \boldsymbol{A} = \boldsymbol{P}_\pi \boldsymbol{A} \boldsymbol{P}_\pi^T$. We denote a pair of nodes $i, j \in V$ as isomorphic in $G$ if exists $\pi \in \mathbb{S}_N$ such that $\pi_i = j$, $\boldsymbol{A} = \boldsymbol{P}_\pi \boldsymbol{A} \boldsymbol{P}_\pi^T$, and $\boldsymbol{F} = \boldsymbol{P}_\pi \boldsymbol{F}$. Node features can be defined by the *graph signal* $f : V \to \mathbb{R}^{F_0}$ as a function that maps a node to an $F_0$-dimensional feature in $\mathbb{R}^{F_0}$. Then the signal of the graph $\boldsymbol{F}$ can be represented by a matrix $\boldsymbol{F} = [\mathbf{f}_1, ..., \mathbf{f}_N]^T \in \mathbb{R}^{N \times F_0}$, where $\mathbf{f}_i \in \mathbb{R}^{F_0}$ are the features of node $i \in V$.

**Random graph model for $G$.** Denote the metric-measure space by $(\mathcal{X}, d, \mu)$, where $\mathcal{X}$ is a set, $d$ is a metric, and $\mu$ is a probability Borel measure. A *graphon* is defined as a mapping $W : \mathcal{X} \times \mathcal{X} \to [0, 1]$ [15, 75]. In what follows we define how the graph $G$ is sampled from the graphon models. The signal definition follows Maskey et al. [46, Definition 2.3] and the edge samples follow Airoldi et al. [3], Lawrence and Hyvärinen [34].

**Definition 1** (Random graph model)**.** *We define* $(W, f)$ *as a* random graph model *for $G$ on $(\mathcal{X}, d, \mu)$ with the graphon $W$ : $\mathcal{X} \times \mathcal{X} \to [0, 1]$ and the metric-space signal $f : \mathcal{X} \to \mathbb{R}^{F_0}$. $f \in L^\infty(\mathcal{X})$ is an essentially bounded measurable function with the essential supremum norm. We obtain $(G, \boldsymbol{F})$ by first sampling $N$ i.i.d. random points $X_1, ..., X_N$ from $\mathcal{X}$ with probability density $\mu$, as the nodes of $G$. Then the edge $(i, j)$ between nodes $i$ and $j$ is sampled with probability $W(X_i, X_j)$, i.e., the adjacency matrix $\boldsymbol{A} = (\boldsymbol{A}_{i,j})_{i,j}$ of $G$ is defined as $\boldsymbol{A}_{i,j} = \mathbb{1}(Z_{i,j} < W(X_i, W_j))$ for $i, j = 1, ..., N$, where $\{Z_{i,j}\}_{i,j=1}^N$ are sampled i.i.d. from Uniform$(0, 1)$. The graph signal $\boldsymbol{F} = [\mathbf{f}_1, ..., \mathbf{f}_N]^T \in \mathbb{R}^{N \times F_0}$ is defined as $\mathbf{f}_i = f(X_i)$. We say $(G, \boldsymbol{F}) \sim (W, f)$. Further, we restrict our attention to graphons $W$ such that there exists a constant $d_{min}$ satisfying the* graphon degree $d_W(x) := \int_\mathcal{X} W(x, y) d\mu(y) \geq d_{min} > 0, \forall x \in \mathcal{X}$.

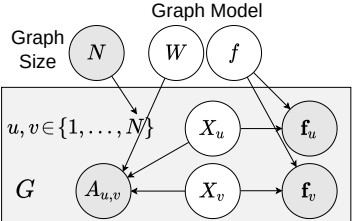

Figure 1: Templated causal DAG of $G$. Hidden and observed variables are shaded white and gray, respectively.

In an abuse of notation we identify node $i \in V$ with the sampled value $X_i \sim \mu, \forall i \in \{1, ..., N\}$, since generally $\mu$ is such that $P(X_i = X_j) = 0$ almost everywhere for $i \neq j$ (e.g., $\mu$ is uniform). The causal DAG of the data generation process of $G$ is given in Figure 1. Our goal is to produce predictors that survive the distribution shift implied by a change in the distribution of graph sizes $N$ during test time. We note that all proofs are relegated to the Appendix due to space constraints.

### 2.1 Inductive structural node representations with graph message-passing neural networks

Henceforth use the terms *node embeddings* and *node representations* interchangeably. Graph message-passing Neural Network (gMPNN$^\bullet$) is defined by realizing a message-passing Neural Network (MPNN) on a graph. We now restate the Maskey et al. [46, Definition 2.1] of MPNN.

**Definition 2** (MPNN [46])**.** *Let $T \in \mathbb{N}$ denote the number of layers. For $t = 1, ..., T$, let $\Phi^{(t)}$ : $\mathbb{R}^{2F_{t-1}} \to \mathbb{R}^{H_{t-1}}$ and $\Psi^{(t)} : \mathbb{R}^{F_{t-1} + H_{t-1}} \to \mathbb{R}^{F_t}$ be functions, where $F_t \in \mathbb{N}$ is called the feature dimension of layer $t$. The corresponding MPNN $\Theta$ is define by the sequence of message functions $(\Phi^{(t)})_{t=1}^T$ and update functions $(\Psi^{(t)})_{t=1}^T$, i.e. $\Theta = ((\Phi^{(t)})_{t=1}^T, (\Psi^{(t)})_{t=1}^T)$.*

We now introduce the gMPNN$^\bullet$ with $T$ message-passing layers. For each node $i \in V$, $\mathbf{f}_i^{\bullet(t)}$ at layer $t \in \{1, ..., T\}$ is defined recursively using (a) its own representation at layer $t - 1$ ($\mathbf{f}_i^{\bullet(t-1)}$) and (b) an aggregated representation of its neighbors $m_i^{(t)}$. Unlike Maskey et al. [46, Definition 2.2] considering *mean aggregation, we consider here the (N-normalized)sum representation* as follows:

**Definition 3** (gMPNN$^\bullet$). *(Adaptation of [46, Definition 2.2] to N-normalized GNNs) Let $(G, \boldsymbol{F})$ be a graph with graph signals as in Definition 1 and $\Theta$ be a MPNN as in Definition 2. For layer $t = 1, ..., T$, define $\Theta_{\boldsymbol{A}}^{\bullet(t)}$ as maps from the input graph $G$ and graph signals $\boldsymbol{F}^{(0)} = \boldsymbol{F} \in \mathbb{R}^{N \times F_0}$ to the features in the $t$-th neural layer by*

$$\Theta_{\boldsymbol{A}}^{\bullet(t)} : \mathbb{R}^{N \times F_0} \to \mathbb{R}^{N \times F_t}, \quad \boldsymbol{F} \mapsto \boldsymbol{F}^{(t)} = (\mathbf{f}_i^{\bullet(t)})_{i=1}^N$$

*where $\boldsymbol{F}^{(t)}$ is defined by the (N-normalized) sum aggregation procedure, $\forall i \in V$, for $\Theta_{\boldsymbol{A}}^{\bullet(t)}$,*

$$m_i^{(t)} := \frac{1}{N} \sum_{j=1}^N A_{i,j} \Phi^{(t)}(\mathbf{f}_i^{\bullet(t-1)}, \mathbf{f}_j^{\bullet(t-1)}), \quad \mathbf{f}_i^{\bullet(t)} := \Psi^{(t)}(\mathbf{f}_i^{\bullet(t-1)}, m_i^{(t)}).$$

Given a gMPNN$^\bullet$, $\Theta_{\boldsymbol{A}}^{\bullet(T)}$, with $T \geq 1$ layers as in Definition 3, their outputs are $\Theta_{\boldsymbol{A}}^{\bullet(T)}(\boldsymbol{F}) \in \mathbb{R}^{N \times F_T}$ for the ($N$-normalized) sum aggregation, and are henceforth denoted as node embedding outputs of the gMPNN$^\bullet$. We denote $\Theta_{\boldsymbol{A}}^{\bullet(T)}(\boldsymbol{F})_i$ as the node embedding for node $i \in V$.

### 2.2 Node embeddings with *continuous* message passing neural networks

Here we adapt the degree-normalized definition of Maskey et al. [46] on continuous message passing for structural node embeddings to our continuous integral aggregation (N-normalized GNNs).

**Definition 4** (Continuous message-passing). *Given a MPNN $\Theta$ as in Definition 2, the* node continuous message passing neural network (cMPNN$^\bullet$) *on graphons is defined by using the metric-space signals $f : \mathcal{X} \to \mathbb{R}^{F_0}$ and continuous aggregation to replace the graph node features and the aggregation scheme in Definition 3. Using a message signal $U : \mathcal{X} \times \mathcal{X} \to \mathbb{R}^H$, the continuous integral aggregation is defined as $M_W^\bullet(U)(x) = \int_{\mathcal{X}} W(x, y) U(x, y) d\mu(y)$, where $W$ is a graphon.*

As defined in Maskey et al. [46, Definition 2.4], the same MPNN $\Theta$ can process metric-space signals instead of graph signals with the continuous aggregations. Instead of using continuous mean aggregation as [46, Definition 2.4], we are using continuous integral aggregation.

**Definition 5** (cMPNN$^\bullet$). *(Adaptation of [46, Definition 2.4] to N-normalized GNNs) Let $(W, f)$ be a random graph model as in Definition 1 and $\Theta$ be a MPNN as in Definition 2. For $t = 1, ..., T$, define $\Theta_W^{\bullet(t)}$ as maps from input metric-space signal $f^{\bullet(0)} = f : \mathcal{X} \to \mathbb{R}^{F_0}$ to the features in the $t$-th layer by*

$$\Theta_W^{\bullet(t)} : L^2(\mathcal{X}) \to L^2(\mathcal{X}), \quad f^\bullet \mapsto f^{\bullet(t)},$$

*where $\overline{f}^{\bullet(t)}$ are defined sequentially aggregation for $\Theta_W^{\bullet(t)}$:*

$$g^{\bullet(t)}(x) := \int_{\mathcal{X}} W(x, y) \Phi^{(t)}(f^{\bullet(t-1)}(x), f^{\bullet(t-1)}(y)) d\mu(y), f^{\bullet(t)}(x) := \Psi^{(t)}(f^{\bullet(t-1)}(x), g^{\bullet(t)}(x)).$$

## 3 Size-stability of node representation and its drawbacks

We now present our results about convergence of gMPNN$^\bullet$ to cMPNN$^\bullet$ for test graphs $G^{\text{te}}$ sampled from the graphon random graph model (see Definition 1), and how it leads to size-stability of gMPNN$^\bullet$ for nodes that have the same representation under cMPNNs$^\bullet$. Theoretical proofs and common definitions (e.g., Lipschitz continuous functions) are relegated to the Appendix to save space.

### 3.1 Convergence of gMPNNs towards cMPNNs as test graph size increase

We now prove that, with high probability, the maximum infinity difference between the gMPNN$^\bullet$ and cMPNN$^\bullet$ node representations decreases with $N^{\text{te}}$, the size of $G^{\text{te}}$. The proof of Theorem 1 closely follows the pointwise convergence proof in Maskey et al. [45], adapted to our OOD setting and can be found in the Appendix.

**Theorem 1** (OOD convergence without in-distribution convergence). *For a random graph model $(W, f)$ satisfying Definition 1, let $N^{tr}$ be a random variable defining the distribution of graph sizes in training. Define the test distribution $(G^{te}, \boldsymbol{F}^{te}) \sim (W, f)$ through the causal graph in Figure 1 as an interventional change to obtain larger test graph sizes where $\min(supp(N^{te})) \gg M_{tr} = \max(supp(N^{tr}))$ (which means any test graph is much larger than the largest possible training graph). Let $\Theta = ((\Phi^{(l)})_{l=1}^T, (\Psi^{(l)})_{l=1}^T)$ be a MPNN as in Definition 2 with $T$ layers such that*

$\Phi^{(l)} : \mathbb{R}^{2F_{l-1}} \to \mathbb{R}^{H_{l-1}}$ and $\Psi^{(l)} : \mathbb{R}^{F_{l-1}+H_{l-1}} \to \mathbb{R}^{F_l}$ are learned from the training distribution and are Lipschitz continuous with Lipschitz constants $L_\Phi^{(l)}(M_{tr})$ and $L_\Psi^{(l)}(M_{tr})$ that depend on $M_{tr}$. Let gMPNN$^\bullet$ $\Theta_A^{\bullet(T)}$ and cMPNN$^\bullet$ $\Theta_W^{\bullet(T)}$ be as in Definitions 3 and 5. Let $X_1^{te}, ..., X_{N^{te}}^{te}$ and $\boldsymbol{A}^{te}$ be as in Definition 1. Let $p \in (0, \frac{1}{\sum_{l=1}^T 2(H_l+1)})$. Then, if

$$\frac{\sqrt{N^{te}}}{\sqrt{\log(2N^{te}/p)}} \geq \frac{4\sqrt{2}}{d_{min}}, \tag{1}$$

we have with probability at least $1 - \sum_{l=1}^T 2(H_l+1)p$,

$$\delta_{A\text{-}W}^\bullet := \max_{i=1,...,N^{te}} \|\Theta_{\boldsymbol{A}^{te}}^{\bullet(T)}(\boldsymbol{F}^{te})_i - \Theta_W^{\bullet(T)}(f)(X_i^{te})\|_\infty \leq (C_1 + C_2\|f\|_\infty)\frac{\sqrt{\log(2N^{te}/p)}}{\sqrt{N^{te}}},$$

where the constants $C_1$ and $C_2$ are defined in the Appendix and depend on $\{L_\Phi^{(l)}(M_{tr}), L_\Psi^{(l)}(M_{tr})\}_{l=1}^T$ and the distribution of $(G^{tr}, \boldsymbol{F}^{tr})$.

Theorem 1 above shows that as the test graph size $N^{te}$ grows, the node representations from the discrete gMPNNs$^\bullet$ learned in the training data converge to the continuous cMPNNs$^\bullet$. *Theorem 1's OOD statement has profound consequences when it comes to predicting links using the node representations obtained by a gMPNN$^\bullet$.* Next, Corollary 1 shows that for any two nodes $i, j \in V^{te}$ that are indistinguishable in the cMPNN$^\bullet$ (defined as $\Theta_W^{\bullet(T)}(f)(X_i^{te}) = \Theta_W^{\bullet(T)}(f)(X_j^{te})$), they will get increasingly similar representations in the discrete gMPNN$^\bullet$ as $N^{te}$ grows.

**Corollary 1.** *Let* $\Theta = ((\Phi^{(l)})_{l=1}^T, (\Psi^{(l)})_{l=1}^T), \Theta_A^{\bullet(T)}, \Theta_W^{\bullet(T)}, p, (W, f), (G^{tr}, \boldsymbol{F}^{tr}), (G^{te}, \boldsymbol{F}^{te}), N^{tr},$ $N^{te}, A^{te},$ *and* $X_1^{te}, ..., X_{N^{te}}^{te}$ *be as in Theorem 1. If there exists* $i, j \in V^{te}, i \neq j$, *s.t.* $\Theta_W^{\bullet(T)}(X_i) = \Theta_W^{\bullet(T)}(X_j)$ *and Equation* (1) *is satisfied, then, with* $C_1$ *and* $C_2$ *as in Theorem 1, we have that with probability at least* $1 - \sum_{l=1}^T 2(H_l+1)p$,

$$\|\Theta_{\boldsymbol{A}^{te}}^{\bullet(T)}(\boldsymbol{F}^{te})_i - \Theta_{\boldsymbol{A}^{te}}^{\bullet(T)}(\boldsymbol{F}^{te})_j\|_\infty \leq (C_1 + C_2\|f\|_\infty)\frac{2\sqrt{\log(2N^{te}/p)}}{\sqrt{N^{te}}}.$$

**Implications of Corollary 1 on Stochastic Block Models (SBMs).** In what follows, we will discuss circumstances where two nodes $i, j \in V$ get the same cMPNN$^\bullet$ representations (i.e., $\Theta_W^{\bullet(T)}(f)(X_i) = \Theta_W^{\bullet(T)}(f)(X_j)$). We will restrict our results to an important family of graphon models: Stochastic Block Models (SBMs) [63], where we also model node attributes. SBMs were chosen because they can consistently model large graphs generated by any piecewise Lipschitz graphon model [3]. SBMs are also intuitive models, which makes them useful to illustrate our results.

**Definition 6** (Stochastic Block Model (SBM)). *An SBM* $(W, f)$ *is a random graph model (Definition 1) with cluster structures in* $W$ *and* $f$. *Partition the node set into* $r \geq 2$ *disjoint subsets* $S_1, S_2, ..., S_r \subseteq V$ *(known as blocks or communities) with an associated* $r \times r$ *symmetric matrix* $\boldsymbol{S}$, *where the probability of an edge* $(i, j)$, $i \in S_a$ *and* $j \in S_b$ *is* $\boldsymbol{S}_{ab}$, *for* $a, b \in \{1, ..., r\}$. *Let* $\mathcal{X} = [0, 1]$, *and* $\mu$ *be the uniform distribution on* $[0, 1]$. *By dividing* $\mathcal{X} = [0, 1]$ *into disjoint convex sets* $[t_0, t_1), [t_1, t_2), ..., [t_{r-1}, t_r]$, *where* $t_0 = 0$ *and* $t_r = 1$, *node* $i$ *belongs to block* $S_a$ *if* $X_i \sim Uniform(0, 1)$ *satisfies* $X_i \in [t_{a-1}, t_a)$. *The graphon function* $W$ *is defined as* $W(X_i, X_j) = \sum_{a,b \in \{1,...,r\}} \boldsymbol{S}_{ab} \mathbb{1}(X_i \in [t_{a-1}, t_a)) \mathbb{1}(X_j \in [t_{b-1}, t_b))$. *We take the liberty to also define node signals in our SBM model, where for* $\boldsymbol{B} = [B_1, ..., B_r]^T \in \mathbb{R}^{r \times F_0}$ *the metric-space signal* $f : \mathcal{X} \to \mathbb{R}^{F_0}$ *is defined as* $f(x) = \sum_{a \in \{1,...,r\}} \mathbb{1}(x \in [t_{a-1}, t_a))B_a$.

We define the action of permutation $\pi$ on $\boldsymbol{B}$ of Definition 6 as $\pi \circ \boldsymbol{B}$, where $(\pi \circ \boldsymbol{B})_{\pi_a} = \boldsymbol{B}_a$.

**Definition 7** (Isomorphic SBM blocks). *For the SBM model* $(W, f)$ *in Definition 6, we say two blocks* $a, b \in \{1, ..., r\}$ *are isomorphic if the SBM satisfies the following two conditions: (a)* $t_a - t_{a-1} = t_b - t_{b-1}$, *and (b) for* $\pi \in \mathbb{S}_r$, *such that* $\pi_a = b$, $\pi_b = a$ *and* $\pi_c = c, \forall c \in \{1, ..., r\}$, $c \neq a, b$, $\boldsymbol{S} = \pi \circ \boldsymbol{S}$, *and* $\boldsymbol{B} = \pi \circ \boldsymbol{B}$.

A similar definition can be obtained for the general graphons in Definition 1 using the isomorphic graphon definition of Lovász and Szegedy [40].

Now that we have the definition for isomorphic blocks in SBM models, we can prove that all nodes in these isomorphic blocks will obtain the same representations under integral aggregation cMPNNs$^\bullet$.

**Lemma 1.** *Let $\Theta = ((\Phi^{(l)})_{l=1}^T, (\Psi^{(l)})_{l=1}^T)$ be a MPNN as in Definition 2, and $\Theta_W^{\bullet(T)}$ as in Definition 5. For the SBM model $(W, f)$ in Definition 6 with $N^{te}$ nodes $X_1, \ldots, X_{N^{te}}$. If there exists $i, j \in V^{te}$ such that $X_i^{te}, X_j^{te}$ are nodes that belong to isomorphic SBM blocks (Definition 7), then $\Theta_W^{\bullet(T)}(f)(X_i^{te}) = \Theta_W^{\bullet(T)}(f)(X_j^{te})$.*

Note that even though any two nodes in isomorphic SBM blocks get the same cMPNN$^\bullet$ representations per Lemma 1, these nodes are likely not isomorphic in $G^{te}$ (as shown in Proposition 1 in Appendix) and, hence, they get different gMPNN$^\bullet$ representations. However, Corollary 1 shows that these representations become increasingly similar as the test graph size grows. We use this observation to understand the ability of gMPNNs$^\bullet$ to perform link prediction tasks next.

### 3.2 The hardness of OOD inductive link prediction using structural node embeddings

The convergence of gMPNNs$^\bullet$ to cMPNNs$^\bullet$ as the test graph size $N^{te}$ grows (Theorem 1) implies through Corollary 1 and Lemma 1 that node representations of distinct SBM blocks can become increasingly similar as the test graph size grows, even though these nodes are not isomorphic in $G^{te}$ with high probability (see Proposition 1 in the Appendix).

**Definition 8** (Link prediction function from structural node embeddings)**.** *An inductive link prediction function $\eta^\bullet : \mathbb{R}^{F_T} \times \mathbb{R}^{F_T} \to [0, 1]$ takes the gMPNN$^\bullet$ node representations of two nodes $i, j \in V^{te}$ and predicts the edge probability $P(A_{ij}^{te} = 1)$. We assume $\eta^\bullet$ is Lipschitz continuous with Lipschitz constant $L_{\eta^\bullet}(M_{tr})$ that depends on $\max(supp(N^{tr}))$. In the context of graphon random graph models (Definition 1), we aim to learn $\eta^\bullet(\Theta_{A^{te}}^{\bullet(T)}(F^{te})_i, \Theta_{A^{te}}^{\bullet(T)}(F^{te})_j) \approx W(i, j)$. We further assume we predict a link if $\eta^\bullet(\cdot, \cdot) > \tau$, while no link if $\eta^\bullet(\cdot, \cdot) < \tau$, for some (arbitrary) threshold $\tau \in [0, 1]$ chosen by the user of such system.*

The next corollary showcases the difficulty in OOD predicting links using structural node representations as $N^{te}$ grows.

**Corollary 2.** *Let $\Theta = ((\Phi^{(l)})_{l=1}^T, (\Psi^{(l)})_{l=1}^T)$ be the MPNN with $T$ layers and $\Theta_A^{\bullet(T)}, \Theta_W^{\bullet(T)}$ as in Theorem 1. Let $\eta^\bullet : \mathbb{R}^{F_T} \times \mathbb{R}^{F_T} \to [0, 1]$ be as in Definition 8. Consider the SBM $(W, f)$ in Definition 6 with isomorphic blocks (Definition 7). Let $(G^{tr}, F^{tr}) \sim (W, f)$ and $(G^{te}, F^{te}) \sim (W, f)$ be the training and test graphs with $N^{tr}$ and $N^{te}$ nodes, respectively as in Theorem 1. Consider any two test nodes $i, j \in \{1, ..., N^{te}\}$, $i \neq j$, for which we can make a link prediction decision with $\eta^\bullet$ (i.e., $\eta^\bullet(\Theta_{A^{te}}^{\bullet(T)}(F^{te})_i, \Theta_A^{\bullet(T)}(F^{te})_j) \neq \tau$). Let $G^{te}$ be large enough to satisfy both Equation (1) and*

$$\frac{\sqrt{N^{te}}}{\sqrt{\log(2N^{te}/p)}} > \frac{2(C_1 + C_2\|f\|_\infty)}{|\eta^\bullet(\Theta_{A^{te}}^{\bullet(T)}(F^{te})_i, \Theta_{A^{te}}^{\bullet(T)}(F^{te})_j) - \tau|/L_\eta^\bullet(M_{tr})},$$

*where $p$, $C_1$, and $C_2$ are as given in Corollary 1. Then, if $i$ and $j$ belong to isomorphic blocks (i.e., $\Theta_W^{\bullet(T)}(f)(X_i^{te}) = \Theta_W^{\bullet(T)}(f)(X_j^{te})$), with probability at least $1 - \sum_{l=1}^T 2(H_l + 1)p$ the link prediction method in Definition 8 will make the same link prediction regardless of the SBM probability matrix $S$ (Definition 6) and whether $i$ and $j$ are in the same block or distinct isomorphic blocks.*

Corollary 2 proves that link prediction with structural node embeddings form gMPNNs$^\bullet$ is unreliable. That is, for any link prediction method satisfying Definition 8, as the test graph grows, the method will increasingly struggle to give different predictions within and across isomorphic SBM blocks, even when these probabilities are arbitrarily different in the underlying graph model. In what follows we show that pairwise embeddings can address this challenge.

## 4 Size-stability of structural *pairwise* embeddings and its advantages

We have discussed the limitation of gMPNNs$^\bullet$ on node representation for link prediction. Now we claim that a joint continuous message passing graph neural network is capable of link prediction in graphon random graph models (Definition 1). We define the joint continuous message passing graph neural network inspired by the cMPNNs$^\bullet$ for node representations (Definition 5). First, we need to define the *graphon fraction of common neighbors* for graphon nodes $x$ and $y$, $c_W(x, y) := \int_\mathcal{X} W(x, z)W(y, z)d\mu(z)$. We only consider graphons $W$ such that there exists $d_{cmin}$ satisfying $c_W(x, y) \geq d_{cmin} > 0, \forall x, y \in \mathcal{X}$ in this section. Since we do not have edge feature as in Definition 1, we define the metric-space pair-wise signal as $f^{\bullet\bullet}(x, y) = 1, \forall x, y \in \mathcal{X}$.

**Definition 9** (cMPNN$^{\bullet\bullet}$). *Let $(W, f)$ be a random graph model as in Definition 1 and $\Theta$ be a MPNN as in Definition 2. For $t = 1, ..., T$, define the continuous (pairwise) cMPNN$^{\bullet\bullet}$ $\Theta_W^{\bullet\bullet(t)}$ as the mapping that maps input pairwise metric-space signals $f^{\bullet\bullet(0)} = f^{\bullet\bullet}$ to the features in the t-th layer by*

$$\Theta_W^{\bullet\bullet(t)} : L^2(\mathcal{X}, \mathcal{X}) \to L^2(\mathcal{X}, \mathcal{X}), \quad f^{\bullet\bullet(0)} \mapsto f^{\bullet\bullet(t)},$$

*where $f^{\bullet\bullet(t)}$ are defined recursively by*

$$g^{\bullet\bullet(t)}(x, y) := \frac{1}{2} \int_\mathcal{X} \left( \frac{W(y, z)}{c_W(x, y)} \Phi^{(t)}(f^{\bullet\bullet(t-1)}(x, y), f^{\bullet\bullet(t-1)}(x, z)) \right.$$
$$\left. + \frac{W(x, z)}{c_W(x, y)} \Phi^{(t)}(f^{\bullet\bullet(t-1)}(x, y), f^{\bullet\bullet(t-1)}(y, z)) \right) d\mu(z),$$
$$f^{\bullet\bullet(t)}(x, y) := \Psi^{(t)}(f^{\bullet\bullet(t-1)}(x, y), g^{\bullet\bullet(t)}(x, y)).$$

The intuition of the aggregation function is that two edges with one same node is considered neighbors in a higher-order graph [51], and to go from $(x, y)$ to $(x, z)$, we need to transition from $y$ to $z$, which has probability $W(y, z)$. The same holds for going from $(x, y)$ to $(y, z)$.

**Lemma 2.** *If $\Phi(x, y) = y$ and $\Psi(x, y) = x/y$, then $f^{\bullet\bullet(t)}(x, y) = W(x, y)$, $\forall x, y \in \mathcal{X}$ is a stationary point in the cMPNN$^{\bullet\bullet}$, i.e. if $f^{\bullet\bullet(t-1)}(x, y) = W(x, y)$, then $f^{\bullet\bullet(t)}(x, y) = W(x, y)$, $\forall x, y \in \mathcal{X}$.*

We define the corresponding gMPNN$^{\bullet\bullet}$ as follows. First we define the *fraction of common neighbors* between nodes $i$ and $j$ as $c_{Ai,j} = \frac{1}{N} \sum_{z=1}^N A_{i,z} \cdot A_{j,z}$. If two nodes do not have common neighbors, then we set $c_{Ai,j} = \frac{1}{N}$ to avoid computation error. Further, we define $\mathbf{f}^{\bullet\bullet}{}_{i,j} = 1 \; \forall i, j \in V$ for any graph $G$, and $\boldsymbol{F}^{\bullet\bullet} = (\mathbf{f}^{\bullet\bullet}{}_{i,j})_{i,j \in V}$ as the pair-wise graph signals.

**Definition 10** (gMPNN$^{\bullet\bullet}$). *Let $(G, \boldsymbol{F})$ be a graph with graph signals as in Definition 1 and $\Theta$ be a MPNN as in Definition 2. For $t = 1, ..., T$ layers we define the gMPNN$^{\bullet\bullet}$ $\Theta_A^{\bullet\bullet(t)}$ as the mapping that maps input pairwise graph signals $\boldsymbol{F}^{\bullet\bullet(0)} = \boldsymbol{F}^{\bullet\bullet}$ to the features in the $t - th$ layer by*

$$\Theta_A^{\bullet\bullet(t)} : \mathbb{R}^{N^2 \times F_0} \to \mathbb{R}^{N^2 \times F_t}, \boldsymbol{F}^{\bullet\bullet(0)} \mapsto \boldsymbol{F}^{\bullet\bullet(t)} = (\mathbf{f}^{\bullet\bullet(t)}_{i,j})_{i,j=1}^N$$

*where $\mathbf{f}^{\bullet\bullet(t)}$ are defined recursively by the following function,*

$$m^{\bullet\bullet(t)}_{i,j} := \frac{1}{2N} \sum_{z=1}^N \frac{A_{j,z}}{c_{Ai,j}} \Phi^{(t)}(\mathbf{f}^{\bullet\bullet(t-1)}_{i,j}, \mathbf{f}^{\bullet\bullet(t-1)}_{i,z}) + \frac{A_{i,z}}{c_{Ai,j}} \Phi^{(t)}(\mathbf{f}^{\bullet\bullet(t-1)}_{i,j}, \mathbf{f}^{\bullet\bullet(t-1)}_{j,z}),$$
$$\mathbf{f}^{\bullet\bullet(t)}_{i,j} := \Psi^{(t)}(\mathbf{f}^{\bullet\bullet(t-1)}_{i,j}, m^{\bullet\bullet(t)}_{i,j}).$$

Next, Theorem 2 shows that these discrete joint representations gMPNN$^{\bullet\bullet}$ converge to the continuous pairwise representation cMPNN$^{\bullet\bullet}$ under the causal DAG of Figure 1.

**Theorem 2** (OOD convergence without in-distribution convergence). *For a random graph model $(W, f)$ satisfying Definition 1, let $N^{tr}$ be a random variable defining the distribution of graph sizes in training. Define the test distribution $(G^{te}, \boldsymbol{F}^{te}) \sim (W, f)$ through the causal graph in Figure 1 as an interventional change to obtain larger test graph sizes where $\min(supp(N^{te})) \gg M_{tr} = \max(supp(N^{tr}))$ (which means any test graph is much larger than the largest possible training graph). Let $\Theta = ((\Phi^{(l)})_{l=1}^T, (\Psi^{(l)})_{l=1}^T)$ be a MPNN as in Definition 2 with $T$ layers such that $\Phi^{(l)}$ and $\Psi^{(l)}$ that are learned from the training data and are Lipschitz continuous with Lipschitz constants $L_\Phi^{(l)}(M_{tr})$ and $L_\Psi^{(l)}(M_{tr})$. Let gMPNN$^{\bullet\bullet}$ $\Theta_W^{\bullet\bullet(T)}$ and cMPNN$^{\bullet\bullet}$ $\Theta_W^{\bullet\bullet(T)}$ be as in Definitions 9 and 10. For a random graph model $(W, f)$ as in Definition 1 with $d_{cmin} > 0$. Let $X_1^{te}, ..., X_{N^{te}}^{te}$ and $\boldsymbol{A}^{te}$ be as in Definition 1. Let $p \in (0, \frac{1}{\sum_{l=1}^T 2(H_l+1)})$. Then, if $\frac{\sqrt{N^{te}}}{\sqrt{\log(2(N^{te})^2/p)}} \geq \frac{4\sqrt{2}}{d_{cmin}}$, we have with probability at least $1 - \sum_{l=1}^T 2(H_l + 1)p$,*

$$\delta_{A\text{-}W}^{\bullet\bullet} = \max_{i,j=1,...,N^{te}} \|\Theta_A^{\bullet\bullet(T)}(\boldsymbol{F}^{\bullet\bullet})_{i,j} - \Theta_W^{\bullet\bullet(T)}(f^{\bullet\bullet})(X_i^{te}, X_j^{te})\|_\infty \leq (C_3 + C_4 \|f^{\bullet\bullet}\|_\infty) \frac{\sqrt{\log(2(N^{te})^2/p)}}{\sqrt{N^{te}}},$$

*where the constants $C_3$ and $C_4$ are defined in the Appendix and depend on $\{L_\Phi^{(l)}(M_{tr}), L_\Psi^{(l)}(M_{tr})\}_{l=1}^T$ and the distribution of $(G^{tr}, \boldsymbol{F}^{tr})$.*

Hence, as the test graph size $N^{\text{te}}$ gets larger w.r.t. $N^{\text{tr}}$ (that is, as we intervene on the causal DAG of Figure 1 to change the support of the distribution of $N$ in order to obtain larger test graphs), the link predictor learned in the training data using gMPNN$^{\bullet\bullet}$ will converge to a continuous method (cMPNN$^{\bullet\bullet}$) that can predict links in OOD tasks (i.e., $W(X_i^{\text{te}}, X_j^{\text{te}})$ is a stationary solution of cMPNN$^{\bullet\bullet}$ per Lemma 2). This convergence is also observed in our empirical results.

## 5 Further Related Work

In what follows we describe works related to learning transferability in GNNs. The concept of transferability of GNN is introduced by Levie et al. [35], Ruiz et al. [58], which state that if two graphs represent same phenomena (e.g., are sampled from the same distribution), then a transferable GNN has approximately the same predictive performance on both graphs. This is closely related to in-distribution generalization capabilities of GNNs to unseen test data, i.e., generalization error when train and test data come from the same distribution. Existing works [26, 44, 58, 59] prove the transferability for spectral-based GCNs under graphon models, and Maskey et al. [46] extends these results to more general message passing GNNs. The GNN smoothness conditions needed to prove uniform convergence of node-embedding equivariant GNNs in Maskey et al. [46] means their GNNs would be unable to perform *in-distribution* link prediction in some tasks (such as the graphs in Definition 7). However, in practice, we observe (Section 6) that GNNs are capable of performing these in-distribution link prediction tasks. Our results are also based on general message passing GNNs. Our goal (OOD link prediction) is, however, significantly different than these prior works, which focus on in-distribution graph and node classification. The link prediction challenge for node-embedding equivariant GNNs is either in symmetric graphs (Srinivasan and Ribeiro [65]) or OOD (this work). Theorem 1 and Corollary 1 say the difference of node representations in isomorphic blocks vanish as the test graphs grow larger, but Theorem 2 says that our pairwise equivariant representation is capable of performing these OOD link prediction task. Related works relating to the representation power, higher order structural and positional link prediction methods (not already covered in our introduction) can be found in Appendix A due to space constraints.

## 6 Empirical Evaluation

In what follows we empirically validate our theoretical results in two parts. We implement all our models in Pytorch Geometric [19] and make it available[1]. Due to space constraints we relegate a detailed description of our experiments to the Appendix.

**Convergence and stability.** First we will empirically validate Theorems 1 and 2 and Corollary 1. Consider an SBM (Definition 6) with three blocks ($r = 3$) and $\boldsymbol{S}_{a,a} = 0.55$, $a = 1, 2, 3$, $\boldsymbol{S}_{1,2} = \boldsymbol{S}_{2,1} = 0.05$, $\boldsymbol{S}_{1,3} = \boldsymbol{S}_{3,1} = 0.02$. Note that one and three are isomorphic blocks (see Definition 7). We use a randomly initialized GraphSAGE [24] GNN model for node embedding, and test both the $\Phi$ and $\Psi$ of Lemma 2, and a scenario where $\Psi$ is a randomly-initialized MLP for pairwise embeddings.

Figures 2(a-c) show log-log plots of the convergence of gMPNNs to their continuous cMPNN counterparts as the test graph size $N^{\text{te}}$ increases. The empirical approximation errors $\delta_{\text{A-W}}^{\bullet}$ (Theorem 1) (Figure 2(a)) and $\delta_{\text{A-W}}^{\bullet\bullet}$ (Theorem 2) are shown as a function of the test graph size $N^{\text{te}} = 2^n$, $n = 5, ..., 13$. The empirical results show agreement with the theory since $\delta_{\text{A-W}}^{\bullet}$ and $\delta_{\text{A-W}}^{\bullet\bullet}$ are bounded above by $O(\sqrt{\log N^{\text{te}}}/\sqrt{N^{\text{te}}})$, which is approximated by the slope $-1/2$ in a log-log plot. Figures 2(d-e) show histograms of the difference between gMPNN$^{\bullet}$ embeddings of different nodes in $G^{\text{te}}$. Let $\Delta_{i,j}^{\bullet} := \overline{\Theta}_A^{\bullet(T)}(\boldsymbol{F})_i - \overline{\Theta}_A^{\bullet(T)}(\boldsymbol{F})_j$ for $i, j \in V^{\text{te}}$, $\Delta_{i,j}^{\bullet} \in \mathbb{R}^{F_T}$ and further define $\Delta_{\text{iso (resp. non-iso)}_{i,j}}^{\bullet} := (\Delta_{i,j}^{\bullet})_{\arg\max_k |(\Delta_{i,j}^{\bullet})_k|}$, where $k \in \{1, \ldots, F_T\}$ is the dimension of the embedding. We use subscript `iso` (resp. `non-iso`) when $i, j \in V^{\text{te}}$ are in isomorphic (resp. non-isomorphic) SBM blocks (Definition 7). As $N^{\text{te}}$ increases, Figure 2(d) shows that embeddings between isomorphic blocks converge, validating Corollary 1, while Figure 2(e) shows that non-isomorphic blocks do not.

**Link prediction performance evaluation with SBMs (in-distribution and OOD).** In what follows we introduce empirical results using a SBM similar in the previous setting. Details can be found in Appendix B.3. We start by sampling the training graph $(G^{\text{tr}}, \boldsymbol{F}^{\text{tr}})$ with $N^{\text{tr}} = 10^3$ nodes. We randomly hide 10% of $E^{\text{tr}}$ from the original graph $G^{\text{tr}}$ for link prediction purpose since the goal of link prediction is to predict possible missing links that is not observed in the original graph. We call

[1] https://github.com/yangzez/OOD-Link-Prediction-Generalization-MPNN

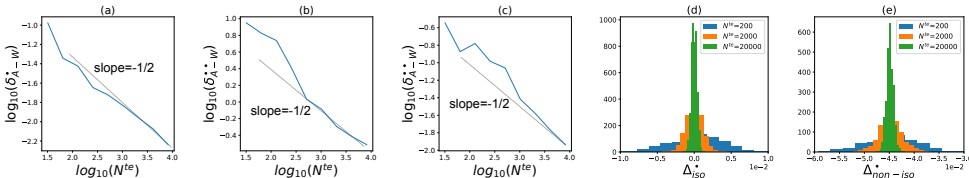

Figure 2: **Experimental agreement with theory:** (a) shows $\delta^{\bullet}_{\text{A-W}}$ (Theorem 1) of a GraphSAGE GNN as a function of $N^{\text{te}}$; (b) shows $\delta^{\bullet\bullet}_{\text{A-W}}$ (Theorem 2) with the gMPNN$^{\bullet\bullet}$ of Lemma 2 as a function of $N^{\text{te}}$; (c) replicates (b) with $\Psi$ as a randomly-initialized neural network. Results shows close agreement with Theorems 1 and 2 that predicts slope $\approx -1/2$ in log-log scale for large $N^{\text{te}}$; (d) shows stable node representations between isomorphic SBM blocks, while (e) shows constant difference in node representations between non-isomorphic SBM blocks, which validate Corollary 1.

these edges $E^{\text{hid-tr}}$. Then we split $E^{\text{hid-tr}}$ into positive train (80%) and validation (10%) edges (we reserve 10% of $E^{\text{hid-tr}}$ for the transductive test scenario), and uniformly sample the same number of across-block non-edges as negative train and validation edges. The embedding method gMPNN$^{\bullet}$ (resp. gMPNN$^{\bullet\bullet}$) along with link predictor $\eta^{\bullet}$ (resp. $\eta^{\bullet\bullet}$) are trained in an end-to-end manner for predicting positive and negative edges in training using cross-entropy loss. Our experiments consider three scenarios (in all scenarios we use the same number of negative test edges as positive test edges, sampled from non-edges in $G^{\text{te}}$ with endpoints in different isomorphic blocks): (i) (In-distribution) transductive scenario where $G^{\text{te}} = G^{\text{tr}}$, where positive test edges are the 10% reserved in $E^{\text{hid-tr}}$ not used in training or validation; (ii) In-distribution inductive scenario where $G^{\text{te}}$ is sampled from the same SBM with $N^{\text{te}} = N^{\text{tr}}$, where we also hide 10% of the edges and sample $0.1|E^{\text{hid-tr}}|$ positive test edges from $E^{\text{hid-te}}$ (for fair comparison across all scenarios); (c) OOD inductive scenario where $G^{\text{te}}$ is sampled from the same SBM with $N^{\text{te}} = 10 \times N^{\text{tr}}$, where we also hide 10% of the edges and sample $0.1|E^{\text{hid-tr}}|$ positive test edges from $E^{\text{hid-te}}$(for fair comparison across all scenarios).

For *structural node embeddings* we consider GraphSAGE [24], GCN [28] (without positional features), GAT [70] and GIN [78] as the representatives of gMPNN$^{\bullet}$ models. The link predictor $\eta^{\bullet}$ is as feedfoward network (with 3 hidden layers and 10 neurons each) that receives the two node embeddings as input, and has link prediction threshold $\tau = 0.5$ (see Definition 8 for details).

For *structural pairwise embeddings* we choose our proposed gMPNN$^{\bullet\bullet}$ method of Definition 10, since we can prove that our approach is theoretically sound in Lemma 2. We test gMPNN$^{\bullet\bullet}$ in two versions: The $\Phi$ and $\Psi$ functions in Lemma 2 (denoted *fixed* $\Psi$) and a feedforward neural network for $\Psi$ with 2 hidden layers and 5 neurons each (denoted *learn* $\Psi$). The link predictor $\eta^{\bullet\bullet}$ is the same as $\eta^{\bullet}$ except it just takes one pairwise embedding as input, rather than two node embeddings.

Table 1 presents our empirical results. The oracle predictor knows the graphon values $W(X_i^{\text{te}}, X_j^{\text{te}})$. Our evaluation metrics include the Matthews correlation coefficient (mcc) [47], balanced accuracy, and Hits@$K$ for $K = 10, 50, 100$ that counts the ratio of positive edges ranked at the $k$-th place or above against all negative edges. Note that gMPNN$^{\bullet}$ structural node representations can very accurately predict links in the transductive tasks, and still performs reasonably well in inductive in-distribution tasks. However, as expected from Corollary 2, this performance suffers significantly as $N^{\text{te}}$ becomes $10\times$ larger than $N^{\text{tr}}$. Now all gMPNN$^{\bullet}$ methods produce predictors that are no better than a random guess over all metrics (e.g., see OOD mcc and accuracy (in red)). In contrast, the gMPNN$^{\bullet\bullet}$ is able to consistently offer good performance on both in-distribution and OOD tasks.

**Link prediction performance evaluation with ogbl-ddi (in-distribution and OOD).** In what follows we introduce empirical results using the ogbl-ddi dataset, which represents a drug-drug interaction network. For the purpose of performing OOD tasks, we start by sampling 10% of the nodes (427 nodes) and its induced subgraph to be the training graph. Further experimental details can be found in Appendix B.4. The in-distribution inductive scenario has $G^{\text{te}}$ constructed as an induced subgraph with $N^{\text{te}} = N^{\text{tr}}$ nodes from the remaining ogbl-ddi graph. Our OOD inductive scenario has $G^{\text{te}}$ as the induced subgraph without the training nodes ($N^{\text{te}} = 3840$ nodes). The test edges are obtained by applying the original edge split on the newly induced test subgraph, where we further down-sample to the same amount of test edges as in our in-distribution scenarios for fair comparison across all scenarios. Table 3 in the Appendix presents our empirical results on the ogbl-ddi link prediction task. All gMPNN$^{\bullet}$ methods performs worse in inductive settings than transductive settings, and suffer much worse performance in OOD transductive setting except GCNs. In contrast, the

Table 1: Test performance over 50 runs of node and pairwise gMPNNs for in-distribution and OOD link prediction over SBM graphs. Methods marked with $*$ indicate best result out of distinct configurations detailed in the Appendix.

| Tasks | | Model | Training graph size $N^{tr} = 10^3$ | | | | |
|---|---|---|---|---|---|---|---|
| | | | Hit@10(%) | Hit@50(%) | Hit@100(%) | mcc.(%) | balanced acc.(%) |
| In-distribution link prediction | Transductive | GraphSAGE* | 95.55( 0.52) | 95.93( 0.73) | 96.14( 0.74) | **95.42( 0.37)** | **97.66( 0.19)** |
| | | GCN* | 93.15(14.57) | 93.99(13.08) | 94.35(12.72) | 92.41(14.72) | 95.97( 8.24) |
| | | GAT* | 93.77(13.03) | 94.01(13.02) | 94.14(13.03) | 90.94(16.09) | 95.26( 8.38) |
| | | GIN* | 95.77( 0.59) | 96.09( 0.58) | 96.28( 0.59) | 95.48( 0.41) | 97.69( 0.22) |
| | | gMPNN•• (fixed $\Psi$) | 93.76( 0.55) | 94.17( 0.51) | 94.51( 0.49) | 93.64( 0.53) | 96.72( 0.28) |
| | | gMPNN•• (learn $\Psi$) | **96.71( 0.32)** | **96.88( 0.31)** | **97.00( 0.30)** | 94.23( 0.55) | 97.03( 0.29) |
| | | **Oracle** | 96.92( 0.36) | 96.92( 0.36) | 96.92( 0.36) | 93.74( 0.42) | 96.77( 0.22) |
| | Inductive $N^{te} = N^{tr}$ | GraphSAGE* | 47.38(39.08) | 52.13(38.87) | 54.94(37.83) | 19.34(43.19) | 61.46(20.17) |
| | | GCN* | 66.29(37.67) | 68.52(35.87) | 69.92(35.12) | 31.76(35.12) | 67.21(22.75) |
| | | GAT* | 40.05(39.05) | 41.34(39.39) | 41.96(39.54) | 19.44(35.22) | 59.52(16.94) |
| | | GIN* | 39.33(34.62) | 42.93(33.86) | 43.90(33.72) | 18.59(39.43) | 59.79(18.24) |
| | | gMPNN•• (fixed $\Psi$) | 93.85( 0.49) | 94.23( 0.51) | 94.55( 0.49) | 93.74( 0.48) | 96.77( 0.25) |
| | | gMPNN•• (learn $\Psi$) | **96.71( 0.30)** | **96.91( 0.28)** | **97.02( 0.27)** | **94.23( 0.59)** | **97.03( 0.31)** |
| | | **Oracle** | 97.01( 0.31) | 97.01( 0.31) | 97.01( 0.31) | 93.87( 0.39) | 96.84( 0.20) |
| OOD link prediction | Inductive $N^{te} = 10^4$ | GraphSAGE* | 9.97(19.47) | 11.73(21.80) | 12.98(23.70) | -6.56( 5.12) | 49.32( 0.60) |
| | | GCN* | 39.29(31.33) | 42.15(30.81) | 44.19(30.97) | -4.88(14.84) | 50.33( 6.72) |
| | | GAT* | 27.31(26.93) | 28.13(26.78) | 28.72(26.93) | -2.00( 8.96) | 50.20( 3.37) |
| | | GIN* | 0.00( 0.00) | 0.00( 0.00) | 0.00( 0.00) | -3.93( 5.12) | 49.59( 0.57) |
| | | gMPNN•• (fixed $\Psi$) | **96.74( 0.07)** | **96.93( 0.04)** | **97.01( 0.04)** | 93.76( 0.05) | 96.78( 0.03) |
| | | gMPNN•• (learn $\Psi$) | **96.97( 0.04)** | **97.02( 0.04)** | **97.08( 0.04)** | **93.94( 0.67)** | **96.88( 0.35)** |
| | | **Oracle** | 96.96( 0.03) | 96.96( 0.03) | 96.96( 0.03) | 93.77( 0.04) | 96.79( 0.02) |

gMPNN•• is able to consistently offer good performance on both in-distribution and OOD tasks, showing that the theoretical results are not limited to SBM models.

# 7    Conclusions

This work studied and provided the first theoretical framework for the task of out-of-distribution (OOD) link prediction, where test graphs are larger than training graphs. Using non-asymptotic bounds, this work showed that OOD link prediction methods using structural node embeddings given by message-passing GNNs converge to link predictors that may perform no better than random guesses. The work also proposed a theoretically-sound structural pairwise embedding with a message-passing algorithm which is able to perform our OOD link prediction task by being approximately invariant to interventions on test graph sizes, as the discrete joint embedding converges to the continuous one. This means that as graph sizes grow in test (OOD), it is still possible to find neural networks parameters that allows our joint representation to converge to the true link probability. We show that the same is not guaranteed for node-embedding equivariant message-passing GNNs. Extensive empirical evaluation showed agreement with these theoretical results. We do not foresee adverse social impacts for this theoretical work, but it does raise awareness of the shortcomings of node-embedding equivariant massage-passing GNNs for link prediction tasks in applications such as recommender systems.

# 8    Acknowledgments

This work was funded in part by the National Science Foundation (NSF) Awards CAREER IIS-1943364 and CCF-1918483, the Purdue Integrative Data Science Initiative, the Purdue Research Foundation, and the Wabash Heartland Innovation Network. Gitta Kutyniok acknowledges support from the ONE Munich Strategy Forum (LMU Munich, TU Munich, and the Bavarian Ministery for Science and Art), the Konrad Zuse School of Excellence in Reliable AI (DAAD), the Munich Center for Machine Learning (BMBF) as well as the German Research Foundation under Grants DFG-SPP-2298, KU 1446/31-1 and KU 1446/32-1 and under Grant DFG-SFB/TR 109, Project C09 and the Federal Ministry of Education and Research under Grant MaGriDo. Any opinions, findings and conclusions or recommendations expressed in this material are those of the authors and do not necessarily reflect the views of the sponsors.

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
