# Appendix of *"OOD Link Prediction Generalization Capabilities of Message-Passing GNNs in Larger Test Graphs"*

In Appendix A, we introduce more related work that has not been discussed in the main paper. In Appendix B, we provide more details in experiments set up and model training. In Appendix C, we introduce notations and definitions that we will use throughout the rest of the appendix. In Appendix D, we show large random and real world graphs have few isomorphic nodes. In Appendix E, we prove the convergence results (Theorem 1) for gMPNN$^{\bullet}$. In Appendix F, we prove the results for hardness of link prediction for gMPNN$^{\bullet}$. Finally, we prove the convergence results for gMPNN$^{\bullet\bullet}$ and cMPNN$^{\bullet\bullet}$ (Theorem 2) in Appendix G.

## A    Further Related Work

**Representation power of GNNs.**    The representation power of GNNs is widely studied in recent years. [51, 78] first show that gMPNN is no more powerful than 1-WL test [73]. Many works have been proposed [43, 51–53] to increase the representation power of GNNs for graph representation, but little has studied on representation power for node and link prediction.

**Structural link prediction.**    Existing link prediction methods assume that, with powerful enough node representations, combining them can guarantee powerful link representations [23, 29]. However, Hu et al. [25] empirically shows that these approaches perform worse than simple heuristic approaches such as Common Neighbor and Adamic-Adar [1, 37]. Theoretically, Srinivasan and Ribeiro [65] was the first work to formally analyze the difference between structural node and link representations, and show that even most-expressive structural node representations are not able to perform link prediction tasks in graphs with high degree of symmetry. In order to remedy this, the state-of-the-art link prediction methods like SEAL [85] use GNNs but transform the task into a graph classification task (the link is an attribute of an induced subgraph around the two target end nodes), where each node in the subgraph are labeled according to their distances to the pair of target end nodes. Zhang et al. [86] unifies such approaches [36, 83, 85] through a method they call "labeling trick", which they show is able to learn structural link representations with a node-most expressive GNN.

**Ability of GNNs to emulate graph algorithms as graph sizes increase.**    Recently, Xu et al. [79] shows that GNNs can extrapolate in algorithmic-related tasks as the graph size grows, if the GNN uses max as an aggregator (rather than sum we considered in this paper). Unfortunately, our Definition 3 of gMPNN$^{\bullet}$ does not allow max aggregators, in part because it is unclear how one could reach stability using the max aggregator. Fortunately, while we could not obtain theoretical results using the max aggregator, we can test it empirically. Table 2 reproduces all our empirical results using the max aggregator (on GraphSAGE and GAT, since these are the only GNNs designed for the max aggregator). Our experiments show that the max aggregator, just like the sum aggregators, shows poor OOD performance as test graph sizes increase. Other works Bevilacqua et al. [7], Yehudai et al. [81] also talk about graph extrapolation as size grows but focus on graph classification. Chen et al. [10], Wu et al. [76, 77] also explore environment-invarian GNN representations for graph classification or node classification tasks. These works differ in that they focus on node classification and graph classification. As Srinivasan and Ribeiro [65] shows, link prediction tasks are significantly different from graph and node classification. Moreover, whether or not one can prove that the max aggregator is or is not able to perform our OOD task is left as future work.

**Positional node embeddings for link prediction.**    Another way to perform link prediction tasks is to use positional node embeddings (PE), which preserves relative positions of the nodes in a graph. The original link predictor in Kipf and Welling [28] uses positional embedding as node attributes for this type of task. However, such approaches can lose the desired permutation equivariance property in graph models. Traditional PE methods include DeepWalk [56] and matrix factorization [2, 49]. You et al. [82] proposes position-aware GNN that only aggregates message from randomly selected anchor nodes, which has poor generalization ability on inductive tasks. Srinivasan and Ribeiro [65] proves that using set representations of PE embeddings over all permutations of the graph input and all random decisions made by the embedding algorithm (e.g., the set of all eigenvectors of randomly permuted graph Laplacian matrices and random eigenvectors due to geometric multiplicity of eigenvalues) can achieve the desired permutation equivariance for link prediction. Dwivedi and

Bresson [16], Kreuzer et al. [33] propose PE that randomly flips of the sign of eigenvectors to alleviate sign ambiguity and pass it to a transformers architecture [68]. Lim et al. [38] proposes a representation that is invariant to the elements of the set described by Srinivasan and Ribeiro [65] in order to achieve equivariant representations for spectral node embeddings. Wang et al. [71] proposes a provable solution for using PEs to learn equivariant and stable representation using separate channels in GNN layers. Dwivedi et al. [17] turns to the idea of learning PE that can be combined with structural representations, and design architecture to decouple structural and positional representations in order to improve both representations.

# B Further Experiment details

In this section we present the details of the experimental section, discussing implementation details. Training was performed on NVIDIA Telsa P100, GeForce RTX 2080 Ti, and TITAN V GPUs.

## B.1 Model implementation

All neural network approaches, including the models proposed in this paper, are implemented in PyTorch [55] and Pytorch Geometric [19] (available respectively under the BSD and MIT license).

Our GIN [78], GCN [28], GAT [70] and GraphSAGE [24] implementations are based on their Pytorch Geometric implementations. We also consider max aggregation as proposed by Xu et al. [80] for extrapolations although it does not fit our theoretical framework.

We use the Adam optimizer to optimize all the neural network models. We use the neural network weights that achieve best validation-set performance for prediction.

## B.2 Empirical validation for convergence and Stability

Consider an SBM (Definition 6) with three blocks ($r = 3$) and $\boldsymbol{S}_{a,a} = 0.55$, $a = 1, 2, 3$, $\boldsymbol{S}_{1,2} = \boldsymbol{S}_{2,1} = 0.05$, $\boldsymbol{S}_{1,3} = \boldsymbol{S}_{3,1} = 0.02$. The probability a node belongs to block one or three is $0.45$, while for block two it is $0.1$. Note that one and three are isomorphic blocks (see Definition 7). Since our results are valid for any gMPNN functions $\Theta$, for our first experiment with node embeddings we use a randomly initialized GraphSAGE [24] GNN model, where following standard GNN procedures we initialize node features as size-normalized degrees (where $d_i = \frac{1}{N} \sum_{j=1,\ldots,N} A_{i,j}$). For the experiment with pairwise embeddings, we test both the $\Phi$ and $\Psi$ of Lemma 2, and a scenario where $\Psi$ is a randomly-initialized feedforward neural network. Later in this section we show how to efficiently compute the exact cMPNN$^\bullet$ and cMPNN$^{\bullet\bullet}$ embeddings of our GraphSAGE and gMPNN$^{\bullet\bullet}$ models.

The validation procedure follows Maskey et al. [45]. We use SBM graphs as examples. Consider an SBM (Definition 6) with three blocks ($r = 3$) and $\boldsymbol{S} = \begin{bmatrix} 0.55 & 0.05 & 0.02 \\ 0.05 & 0.55 & 0.05 \\ 0.02 & 0.05 & 0.55 \end{bmatrix}$. The probability a node belongs to block one or three is $0.45$, while for block two it is $0.1$. The in-block edge probability is $0.55$, and across-isomorphic block probability is $0.02$ and across-non-isomorphic block probability is $0.05$. Note that blocks one and three are isomorphic blocks (see Definition 7).

Since our results is valid for any gMPNN functions, we use a randomly initialized GraphSAGE [24] GNN model for our first experiments with node embeddings. Following our Definition 6, the initial node embeddings within the same block should be the same, however, following standard GNN procedures we initialize node features as size-normalized degrees. Note that in theory, node within each block has the same expected graphon degree, but this setting is more realistic and shows a stronger results than proposed in our theorems when initial node embeddings also have variance.

To efficiently calculate exact cMPNN$^\bullet$ embeddings, we need to make use of the property of SBMs, i.e. the graphon values within a block is constant. The graphon degree $d_W$ for nodes in block 1, 2 and 3 is $0.2615, 0.1$ and $0.2615$. Then we can write the integral $\int_{\mathcal{X}} \frac{W(x,y)}{d_W(x)} \Phi^{(t)}(\bar{f}^{\bullet(t-1)}(x), \bar{f}^{\bullet(t-1)}(y)) d\mu(y)$ as $\frac{1}{0.2615}(0.45 \times \boldsymbol{S}_{1,1} \Phi^{(t)}(\boldsymbol{B}_1^{(t-1)}, \boldsymbol{B}_1^{(t-1)}) + 0.1 \times \boldsymbol{S}_{1,2} \Phi^{(t)}(\boldsymbol{B}_1^{(t-1)}, \boldsymbol{B}_2^{(t-1)}) + 0.45 \times \boldsymbol{S}_{1,3} \Phi^{(t)}(\boldsymbol{B}_1^{(t-1)}, \boldsymbol{B}_3^{(t-1)}))$. This can be calculated exactly by extracting the neural network weights from the GNN model for $\Phi$ and $\Psi$.

Table 2: Test performance over 50 runs of node and pairwise gMPNNs for in-distribution and OOD link prediction over SBM graphs. Methods marked with $*$ indicate best result out of distinct configurations detailed in the Appendix.

| Tasks | | Model | Training graph size $N^{\text{tr}} = 10^3$ | | | | |
|---|---|---|---|---|---|---|---|
| | | | Hit@10(%) | Hit@50(%) | Hit@100(%) | mcc.(%) | balanced acc.(%) |
| In-distribution link prediction | Transductive | GraphSAGE* | 95.55( 0.52) | 95.93( 0.73) | 96.14( 0.74) | **95.42( 0.37)** | **97.66( 0.19)** |
| | | GraphSAGE(max)* | 95.43( 0.38) | 96.13( 0.57) | 96.54( 0.60) | **95.38( 0.36)** | **97.64( 0.19)** |
| | | GCN* | 93.15(14.57) | 93.99(13.08) | 94.35(12.72) | 92.41(14.72) | 95.97( 8.24) |
| | | GAT* | 93.77(13.03) | 94.01(13.02) | 94.14(13.03) | 90.94(16.09) | 95.26( 8.38) |
| | | GAT(max)* | 92.91(12.27) | 93.88( 9.12) | 94.08( 8.82) | 87.36(20.41) | 93.34(10.95) |
| | | GIN* | 95.77( 0.59) | 96.09( 0.58) | 96.28( 0.59) | 95.48( 0.41) | 97.69( 0.22) |
| | | **gMPNN$^{\bullet\bullet}$ (fixed $\Psi$)** | 93.76( 0.55) | 94.17( 0.51) | 94.51( 0.49) | 93.64( 0.53) | 96.72( 0.28) |
| | | **gMPNN$^{\bullet\bullet}$ (learn $\Psi$)** | **96.71( 0.32)** | **96.88( 0.31)** | **97.00( 0.30)** | 94.23( 0.55) | 97.03( 0.29) |
| | | **Oracle** | 96.92( 0.36) | 96.92( 0.36) | 96.92( 0.36) | 93.74( 0.42) | 96.77( 0.22) |
| | Inductive $N^{\text{te}} = N^{\text{tr}}$ | GraphSAGE* | 47.38(39.08) | 52.13(38.87) | 54.94(37.83) | 19.34(43.19) | 61.46(20.17) |
| | | GraphSAGE(max)* | 17.72(22.89) | 25.91(27.75) | 31.43(30.18) | 18.24(30.43) | 58.65(14.53) |
| | | GCN* | 66.29(37.67) | 68.52(35.87) | 69.92(35.12) | 31.76(35.12) | 67.21(22.75) |
| | | GAT* | 40.05(39.05) | 41.34(39.39) | 41.96(39.54) | 19.44(35.22) | 59.52(16.94) |
| | | GAT(max)* | 41.98(39.23) | 43.34(38.71) | 43.54(38.69) | 22.66(38.99) | 61.46(19.02) |
| | | GIN* | 39.33(34.62) | 42.93(33.86) | 43.90(33.72) | 18.59(39.43) | 59.79(18.24) |
| | | **gMPNN$^{\bullet\bullet}$ (fixed $\Psi$)** | 93.85( 0.49) | 94.23( 0.51) | 94.55( 0.49) | 93.74( 0.48) | 96.77( 0.25) |
| | | **gMPNN$^{\bullet\bullet}$ (learn $\Psi$)** | **96.71( 0.30)** | **96.91( 0.28)** | **97.02( 0.27)** | **94.23( 0.59)** | **97.03( 0.31)** |
| | | **Oracle** | 97.01( 0.31) | 97.01( 0.31) | 97.01( 0.31) | 93.87( 0.39) | 96.84( 0.20) |
| OOD link prediction | Inductive $N^{\text{te}} = 10^4$ | GraphSAGE* | 9.97(19.47) | 11.73(21.80) | 12.98(23.70) | -6.56( 5.12) | 49.32( 0.60) |
| | | GraphSAGE(max)* | 1.44( 2.35) | 2.60( 4.76) | 3.58( 6.53) | -2.52( 4.44) | 49.83( 0.57) |
| | | GCN* | 39.29(31.33) | 42.15(30.81) | 44.19(30.97) | -4.88(14.84) | 50.33( 6.72) |
| | | GAT* | 27.31(26.93) | 28.13(26.78) | 28.72(26.93) | -2.00( 8.96) | 50.20( 3.37) |
| | | GAT(max)* | 32.56(26.94) | 33.01(27.16) | 33.24(27.27) | -2.85( 9.76) | 49.82( 3.43) |
| | | GIN* | 0.00( 0.00) | 0.00( 0.00) | 0.00( 0.00) | -3.93( 5.12) | 49.59( 0.57) |
| | | **gMPNN$^{\bullet\bullet}$ (fixed $\Psi$)** | **96.74( 0.07)** | **96.93( 0.04)** | **97.01( 0.04)** | 93.76( 0.05) | **96.78( 0.03)** |
| | | **gMPNN$^{\bullet\bullet}$ (learn $\Psi$)** | **96.97( 0.04)** | **97.02( 0.04)** | **97.08( 0.04)** | **93.94( 0.67)** | **96.88( 0.35)** |
| | | **Oracle** | 96.96( 0.03) | 96.96( 0.03) | 96.96( 0.03) | 93.77( 0.04) | 96.79( 0.02) |

Then we compare the difference between gMPNN$^{\bullet}$ and cMPNN$^{\bullet}$ for increasing number of nodes. We first plot log-log plots, where a $O(\frac{1}{\sqrt{N}})$ decay rate will have slope $-\frac{1}{2}$ in the log-log plot. Our theory bounds the decay rate by $O(\frac{\log N}{\sqrt{N}})$, which can be approximated by the $-\frac{1}{2}$ slope and is validated in Figure 2.

**Pairwise embeddings** For the experiment with pairwise embeddings, we test both the $\Phi$ and $\Psi$ of Lemma 2, and a scenario where $\Psi$ is a randomly initialized two layer feed-forward neural network. To compute the cMPNN$^{\bullet\bullet}$ embeddings, without choosing the adjacency matrix as input to the model, we can input the graphon value matrix $\boldsymbol{W}$ where $\boldsymbol{W}_{i,j} = W(X_i, X_j)$. In our experiment, we choose graph with 20 nodes, 9 in block 1, 2 in block 2, and 9 in block 3. The result of cMPNN$^{\bullet\bullet}$ is stable for graphs with different sizes. Then we plot the same log-log plot as above.

## B.3 Link prediction performance evaluation with SBMs

First, we use a slightly modified SBM with $\boldsymbol{S} = \begin{bmatrix} 0.6 & 0.05 & 0.02 \\ 0.05 & 0.6 & 0.05 \\ 0.02 & 0.05 & 0.6 \end{bmatrix}$ with other things the same as in the above subsection. Here we increase the in-block edge probability to $0.6$ since we are going to hide edges for link prediction purpose.

We start by sampling the training graph $(G^{\text{tr}}, \boldsymbol{F}^{\text{tr}})$ with $N^{\text{tr}} = 10^3$ nodes. We randomly hide 10% of $E^{\text{tr}}$ from the original graph $G^{\text{tr}}$ for link prediction purpose since the goal of link prediction is to predict possible missing links that is not observed in the original graph. We call these edges $E^{\text{hid-tr}}$.

Then we split $E^{\text{hid-tr}}$ into positive train (80%) and validation (10%) edges (we reserve 10% of $E^{\text{hid-tr}}$ for the transductive test scenario), and uniformly sample the same number of across-block non-edges

Table 3: Test performance over 50 runs of node and pairwise gMPNNs for in-distribution and OOD link prediction over the ogbl-ddi graph. Methods marked with ∗ indicate best result out of distinct configurations detailed in the Appendix.

| | | Training graph size $N^{tr} = 427$ | | | | |
|---|---|---|---|---|---|---|
| Tasks | Model | Hit@10(%) | Hit@50(%) | Hit@100(%) | mcc.(%) | balanced acc.(%) |
| In-distribution link prediction — Transductive | GraphSAGE* | 30.23(2.03) | 47.70(1.75) | 60.36(1.79) | 71.47(0.70) | **85.72(0.36)** |
| | GCN* | 17.91(0.52) | 33.69(0.60) | 44.34(0.85) | 59.45(0.50) | 78.85(0.36) |
| | GAT* | 1.46(0.52) | 8.20(1.34) | 16.37(1.95) | 52.64(1.62) | 74.75(0.61) |
| | GIN* | 17.21(4.74) | 28.76(5.79) | 37.46(6.60) | 54.27(1.59) | 76.84(1.19) |
| | gMPNN•• (fixed Ψ) | 14.09(0.06) | 50.32(0.01) | 65.41(0.01) | **73.23(0.10)** | **86.60(0.04)** |
| | gMPNN•• (learn Ψ) | **38.60(1.68)** | **59.04(0.22)** | **68.63(0.06)** | 71.96(0.06) | 85.74(0.03) |
| | **Random** | 0.48(2.58) | 1.16(4.58) | 2.01(6.54) | 0.05(0.39) | 50.00(0.01) |
| In-distribution link prediction — Inductive $N^{te} = N^{tr}$ | GraphSAGE* | 10.52(1.33) | 23.85(1.29) | 36.60(1.37) | 47.58(2.98) | 71.59(2.46) |
| | GCN* | 10.76(0.90) | 24.79(0.73) | 34.99(0.70) | 50.82(0.19) | 74.73(0.21) |
| | GAT* | 0.07(0.02) | 0.22(0.10) | 0.51(0.07) | -0.93(0.77) | 50.00(0.01) |
| | GIN* | 10.95(4.19) | 24.42(5.75) | 33.71(6.70) | 40.67(2.36) | 66.24(1.75) |
| | gMPNN•• (fixed Ψ) | 34.24(0.07) | 66.87(0.03) | 73.91(0.02) | **67.89(0.34)** | **83.76(0.20)** |
| | gMPNN•• (learn Ψ) | **56.45(0.08)** | **68.42(0.03)** | **74.93(0.02)** | 65.55(0.15) | 82.62(0.09) |
| | **Random** | 0.41(1.64) | 2.20(4.88) | 4.97(8.77) | -0.03(0.22) | 50.00(0.00) |
| OOD link prediction — Inductive $N^{te} = 3840$ | GraphSAGE* | 1.79(1.21) | 13.70(6.71) | 25.31(8.77) | 16.65(3.31) | 52.79(1.01) |
| | GCN* | 12.38(1.23) | 27.28(1.27) | 37.45(1.43) | 55.03(0.76) | 77.38(0.36) |
| | GAT* | 2.76(1.27) | 7.55(3.28) | 12.78(4.50) | 23.83(16.31) | 59.54(6.96) |
| | GIN* | 0.00(0.00) | 0.00(0.00) | 0.00(0.00) | 45.87(3.55) | 68.92(2.78) |
| | gMPNN•• (fixed Ψ) | 9.31(5.23) | 67.42(0.02) | 78.44(0.01) | **75.42(0.17)** | **87.37(0.11)** |
| | gMPNN•• (learn Ψ) | **57.97(0.02)** | **74.75(0.07)** | **80.00(0.11)** | 72.04(0.20) | 84.57(0.14) |
| | **Random** | 1.21(3.50) | 3.39(7.72) | 5.71(11.13) | 0.00(0.00) | 50.00(0.00) |

Table 4: Test performance over 50 runs of node and pairwise gMPNNs for in-distribution (large) and OOD (small) link prediction over SBM graphs. Methods marked with ∗ indicate best result out of distinct configurations detailed in the Appendix.

| | | Training graph size $N^{tr} = 10^4$ | | | | |
|---|---|---|---|---|---|---|
| Tasks | Model | Hit@10(%) | Hit@50(%) | Hit@100(%) | mcc.(%) | balanced acc.(%) |
| In-distribution link prediction — Transductive | GraphSAGE* | 75.35(38.50) | 75.41(38.53) | 75.46(38.55) | 70.81(43.47) | 85.93(20.62) |
| | GCN* | 86.23(27.88) | 86.48(27.85) | 86.56(27.85) | 82.73(32.32) | 91.36(15.84) |
| | GAT* | 59.21(43.07) | 59.62(43.09) | 59.79(43.12) | 50.19(42.84) | 75.51(21.35) |
| | GIN* | 80.89(33.65) | 81.12(33.71) | 81.20(33.72) | 82.46(30.01) | 90.49(16.59) |
| | gMPNN•• (fixed Ψ) | 95.74( 0.12) | 96.15( 0.06) | 96.33( 0.04) | 93.77( 0.04) | 96.79( 0.02) |
| | gMPNN•• (learn Ψ) | **96.95( 0.03)** | **96.95( 0.03)** | **96.95( 0.03)** | **93.76( 0.06)** | **96.79( 0.03)** |
| | **Oracle** | 96.96( 0.03) | 96.96( 0.03) | 96.96( 0.03) | 93.77( 0.04) | 96.79( 0.02) |
| In-distribution link prediction — Inductive $N^{te} = N^{tr}$ | GraphSAGE* | 64.77(40.22) | 65.88(39.91) | 66.60(39.87) | 33.19(50.16) | 68.30(23.45) |
| | GCN* | 79.67(34.82) | 79.90(34.55) | 80.07(34.31) | 51.16(49.53) | 76.23(23.72) |
| | GAT* | 46.73(37.62) | 47.12(37.64) | 47.31(37.65) | 19.14(39.03) | 60.02(18.77) |
| | GIN* | 59.68(41.62) | 61.15(41.47) | 61.69(41.37) | 44.80(46.15) | 71.90(22.57) |
| | gMPNN•• (fixed Ψ) | 95.67( 0.11) | 96.15( 0.06) | 96.33( 0.04) | 93.77( 0.05) | 96.79( 0.03) |
| | gMPNN•• (learn Ψ) | **96.94( 0.04)** | **96.94( 0.04)** | **96.94( 0.04)** | **93.76( 0.06)** | **96.78( 0.03)** |
| | **Oracle** | 96.95( 0.04) | 96.95( 0.04) | 96.95( 0.04) | 93.77( 0.05) | 96.79( 0.03) |
| OOD link prediction — Inductive $N^{te} = 10^3$ | GraphSAGE* | 33.52(44.93) | 33.70(44.87) | 33.97(44.77) | 32.72(47.00) | 66.97(22.73) |
| | GCN* | 72.28(40.06) | 73.95(38.58) | 74.17(38.56) | 68.93(40.98) | 84.54(19.69) |
| | GAT* | 23.31(39.07) | 23.32(39.07) | 23.34(39.07) | 24.07(39.18) | 61.74(19.39) |
| | GIN* | 1.31( 1.62) | 1.39( 1.62) | 1.42( 1.62) | -0.64( 5.63) | 49.93( 0.63) |
| | gMPNN•• (fixed Ψ) | 93.68( 0.40) | 93.72( 0.41) | 93.74( 0.41) | 93.40( 0.42) | 96.59( 0.22) |
| | gMPNN•• (learn Ψ) | **96.12( 0.28)** | **96.44( 0.34)** | **96.57( 0.36)** | **94.43( 0.31)** | **97.14( 0.16)** |
| | **Oracle** | 96.94( 0.30) | 96.94( 0.30) | 96.94( 0.30) | 93.82( 0.40) | 96.81( 0.21) |

as negative train and validation edges. The embedding method gMPNN• (resp. gMPNN••) along with link predictor $\eta$• (resp. $\eta$••) are trained in an end-to-end manner for predicting positive and negative edges in training using cross-entropy loss. Our experiments consider three scenarios (in all scenarios we use the same number of negative test edges as positive test edges, sampled from non-edges in $G^{te}$ with endpoints in different isomorphic blocks): (i) (In-distribution) transductive scenario where $G^{te} = G^{tr}$, where positive test edges are the 10% reserved in $E^{hid-tr}$ not used in training or validation; (ii) In-distribution inductive scenario where $G^{te}$ is sampled from the same SBM with $N^{te} = N^{tr}$, where we also hide 10% of the edges and sample $0.1|E^{hid-tr}|$ positive test edges from $E^{hid-te}$ (for fair comparison across all scenarios); (c) OOD inductive scenario where $G^{te}$ is

sampled from the same SBM with $N^{\text{te}} = 10 \times N^{\text{tr}}$, where we also hide $10\%$ of the edges and sample $0.1|E^{\text{hid-tr}}|$ positive test edges from $E^{\text{hid-te}}$ (for fair comparison across all scenarios).

For *structural node embeddings* we consider GraphSAGE [24], GCN [28] (without positional features), GAT [70] and GIN [78] as the representatives of gMPNN$^{\bullet}$ models. Here we also add max aggregation for GAT and GraphSAGE model as proposed by Xu et al. [80] for extrapolation. The link predictor $\eta^{\bullet}$ is as feedfoward network that receives the two node embeddings as input, and has link prediction threshold $\tau = 0.5$ (see Definition 8 for details). We initialize the node features as the size-normalized degrees.

For *structural pairwise embeddings* we choose our proposed gMPNN$^{\bullet\bullet}$ method of Definition 10, since we can prove that our approach is theoretically sound in Lemma 2. We test gMPNN$^{\bullet\bullet}$ in two versions: The $\Phi$ and $\Psi$ functions in Lemma 2 (denoted *fixed* $\Psi$) and a feedforward neural network for $\Psi$ (denoted *learn* $\Psi$). The link predictor $\eta^{\bullet\bullet}$ is the same as $\eta^{\bullet}$ except it just takes one pairwise embedding as input, rather than two node embeddings. We initialize the pairwise features as all 1's to contain no additional information about connectivity between the pair of the nodes.

Many existing link prediction methods rely on positional node embeddings and our work focuses on permutation-equivariant MPNN GNNs. These positional node embedding link prediction methods are not equivariant (they are positional node representations) based on matrix and tensor factorization methods. Developing a theory for the effect of positional node representations in OOD link prediction is far from trivial and an entirely new paper that requires a new theory. At this point we do not even know how positional representations could be approximately counterfactually-invariant.

For all models including gMPNN$^{\bullet}$, gMPNN$^{\bullet\bullet}$, $\eta^{\bullet}$ and $\eta^{\bullet\bullet}$. The number of hidden layers was chosen between $\{2, 3\}$, and the number of hidden neurons was chosen between $\{5, 10\}$ due to the simple experimental set up. For GAT, we have 2 attention heads. Specifically. We optimized all models using Adam with learning rate chosen from $\{1 \times 10^{-3}, 5 \times 10^{-4}, 1 \times 10^{-4}\}$. We also choose $\eta^{\bullet}$ as taking the inner product between pair of nodes as input (as Hu et al. [25]) and the concatenated node embeddings as input. The hyperparameter search is performed by training all models with 10 different initialization seeds and selecting the configuration that achieved the highest mean accuracy on the validation data, and we mark the methods with $*$ in Tables 1 and 2 indicating the optimal configuration is being used. The training time is around 10 minutes for $1,000$ epochs.

Table 2 presents our empirical results in the new setting over 50 independent runs. The oracle predictor knows the graphon values $W(X_i^{\text{te}}, X_j^{\text{te}})$. The reason why it can not achieve $100\%$ accuracy is because there exists rarely sampled positive edges between blocks. Our evaluation metrics include the Matthews correlation coefficient (mcc) [47], balanced accuracy, and Hits@$K$ for $K = 10, 50, 100$ that counts the ratio of positive edges ranked at the $k$-th place or above against all negative edges. The results from the new table conveys the same message as Table 1 and has been discussed in Section 6.

We also include a new setting for training on larger graphs ($10^4$ nodes) and extrapolating to smaller graphs ($10^3$ nodes) in Table 4. We are able to see the structure node representations gMPNN$^{\bullet}$ are still able to perform relatively well on in-distribution inductive tasks, although the graphs are large, while still suffer from OOD performance to smaller graphs except GCN, although it is not related to the theoretical discussions of this paper. In contrast, the gMPNN$^{\bullet\bullet}$ is able to consistently offer good performance on both in-distribution and OOD tasks.

As discussed in Appendix A, Xu et al. [79] shows that GNNs can extrapolate in algorithmic-related tasks as the graph size grows, if the GNN uses max as an aggregator (rather than sum we considered in this paper). Unfortunately, our Definition 3 of gMPNN$^{\bullet}$ does not allow max aggregators, in part because it is unclear how one could reach stability using the max aggregator. Fortunately, while we could not obtain theoretical results using the max aggregator, we can test it empirically. Table 2 reproduces all our empirical results using the max aggregator (on GraphSAGE and GAT, since these are the only GNNs designed for the max aggregator). Our experiments show that the max aggregator, just like the sum aggregators, shows poor OOD performance as test graph sizes increase. Whether there is theoretical proof that the max aggregator is not able to perform this OOD task is left as future work.

## B.4 Link prediction performance evaluation with ogbl-ddi

In what follows we introduce empirical results using the ogbl-ddi dataset, which represents a drug-drug interaction network. For the purpose of performing OOD tasks, we start by sampling $10\%$ of the nodes ($427$ nodes) and its induced subgraph to be the training graph, where node features are constructed as size-normalized degrees in the training graph. Validation positive and negative edges are obtained by applying the original edge split on the induced training subgraph. Our experiments consider three scenarios: (i) (In-distribution) transductive scenario where $G^{\text{te}} = G^{\text{tr}}$, where test positive and negative edges are obtained by applying the original edge split on the induced training subgraph; (ii) In-distribution inductive scenario where $G^{\text{te}}$ is constructed as sampling $N^{\text{te}} = N^{\text{tr}}$ nodes from the remaining ogbl-ddi graph and its induced subgraph, where the test edges are obtained by applying the original edge split on the newly induced test subgraph; (iii) OOD inductive scenario where $G^{\text{te}}$ is the induced subgraph without the training nodes with $N^{\text{te}} = 3840$, the test edges are obtained by applying the original edge split on the newly induced test subgraph, where we further down-sample to the same amount of test edges as in (ii) for fair comparison across all scenarios.

We used the same benchmarking methods as in the SBM experiments, and add a random guesser where it is constructed as randomly-initialized GraphSAGE model with a randomly-initialized link predictor. We initialize the pairwise features as all 1's to contain no additional information about connectivity between the pair of the nodes for gMPNN$^{\bullet\bullet}$.

For *structural node embeddings* we consider GraphSAGE [24], GCN [28] (without positional features), GAT [70] and GIN [78] as the representatives of gMPNN$^{\bullet}$ models. The link predictor $\eta^{\bullet}$ is as feedfoward network that receives the two node embeddings as input, and has link prediction threshold $\tau = 0.5$ (see Definition 8 for details). We initialize the node features as the size-normalized degrees.

For *structural pairwise embeddings* we choose our proposed gMPNN$^{\bullet\bullet}$ method of Definition 10, since we can prove that our approach is theoretically sound in Lemma 2. We test gMPNN$^{\bullet\bullet}$ in two versions: The $\Phi$ and $\Psi$ functions in Lemma 2 (denoted *fixed* $\Psi$) and a feedforward neural network for $\Psi$ (denoted *learn* $\Psi$). The link predictor $\eta^{\bullet\bullet}$ is the same as $\eta^{\bullet}$ except it just takes one pairwise embedding as input, rather than two node embeddings. We initialize the pairwise features as all 1's to contain no additional information about connectivity between the pair of the nodes.

For all models including gMPNN$^{\bullet}$, gMPNN$^{\bullet\bullet}$, $\eta^{\bullet}$ and $\eta^{\bullet\bullet}$. The number of hidden layers was chosen between $\{2, 3\}$, and the number of hidden neurons was chosen between $\{16, 32\}$ due to the simple experimental set up. For GAT, we have 2 attention heads. Specifically. We optimized all models using Adam with learning rate chosen from $\{1 \times 10^{-3}, 5 \times 10^{-4}, 1 \times 10^{-4}\}$. We also choose $\eta^{\bullet}$ as taking the inner product between pair of nodes as input (as Hu et al. [25]) and the concatenated node embeddings as input. We train all the models with 200 epochs. The hyperparameter search is performed by training all models with 10 different initialization seeds and selecting the configuration that achieved the highest mean accuracy on the validation data, and we mark the methods with $*$ in Table 3 indicating the optimal configuration is being used.

Table 3 presents our empirical results on the ogbl-ddi link prediction task. All gMPNN$^{\bullet}$ methods performs worse in inductive settings than transductive settings, and suffer much worse performance in OOD transductive setting except GCNs. In contrast, the gMPNN$^{\bullet\bullet}$ is able to consistently offer good performance on both in-distribution and OOD tasks, showing that the theoretical results are not limited to SBM models.

## C  Definition and notations

In this section, we follow the definitions and notations from Maskey et al. [46, Appendix A]. As in Maskey et al. [46, Appendix A], we call the metric space $(\chi, d)$, where the metric in the space $\chi$ is defined as $d : \chi \times \chi \to [0, \infty)$. The nodes of the graph are considered as sampled point from $\chi$, the node $i$ is identified with $X_i$ for the graph $G$ with nodes $X = (X_1, \ldots, X_N)$. We also represent $\boldsymbol{F}(X_i) := \mathbf{f}_i$ for $i = 1, \ldots, N$.

Next, we define various notions of degree for the pairwise node embedding.

**Definition 11.** *Let $W$ defined in Definition 1, and $G$ as the sampled graph with nodes $X = (X_1, ..., X_N)$.*

- *We define the graphon fraction of common neighbors at $x, y \in \mathcal{X}$ by*

$$c_W(x, y) = \int_{\mathcal{X}} W(x, z)W(y, z)d\mu(z), \tag{2}$$

- *Given two points $x, y$ that need not be in $X$, we define the graph-graphon fraction of common neighbors of $X$ at $x, y$ by*

$$c_X(x, y) = \frac{1}{N} \sum_{i=1}^{N} W(x, X_i)W(y, X_i), \tag{3}$$

- *Given two points $x, y$ that need not be in $X$, we define the sampled-graph fraction of common neighbors of $X$ at $x, y$ by*

$$c_A(x, y) = \frac{1}{N} \sum_{i=1}^{N} A(x, X_i)A(y, X_i), \tag{4}$$

*where we define $A(x, X_i) \sim Ber(W(x, X_i))$ and $A(y, X_i) \sim Ber(W(y, X_i))$ as independent random variables.*

where $c_X(x, y)$ and $c_A(x, y)$ are interpreted as the graph fraction of common of neighbors of the node pair $(x, y)$ in the graph $(x, y, X_1, ..., X_n)$.

Adapting Maskey et al. [46, Definition A.3] to the continuous integral aggregation,

**Definition 12.** *Let $W$ be defined in Definition 1, for a metric-space message signal $U : \mathcal{X} \times \mathcal{X} \to \mathbb{R}^F$, the continuous integral aggregation is defined by*

$$M_W^{\bullet} U = \int_{\mathcal{X}} W(\cdot, y)U(\cdot, y)d\mu(y).$$

Adapting Maskey et al. [46, Definition A.4] to the N-normalized sum aggregation,

**Definition 13.** *Let $W$ be defined in Definition 1, $X = X_1, ..., X_n$ sample points. For a metric-space message signal $U : \mathcal{X} \times \mathcal{X} \to \mathbb{R}^F$, we define the graph-graphon (N-normalized) sum aggregation by*

$$M_X^{\bullet} U = \frac{1}{N} \sum_i W(\cdot, X_i)U(\cdot, X_i),$$

*and the sampled-graph (N-normalized) sum aggregation by*

$$M_A^{\bullet} U = \frac{1}{N} \sum_i A(\cdot, X_i)U(\cdot, X_i),$$

*where we define $A(x, X_i) \sim Ber(W(x, X_i))$ as a random variable.*

**Definition 14.** *Let $W$ be defined in Definition 1, $X = X_1, ..., X_n$ sample points. For a metric-space message signal $U : (\mathcal{X} \times \mathcal{X}) \times (\mathcal{X} \times \mathcal{X}) \to \mathbb{R}^F$, we define the graphon pairwise aggregation by*

$$M_W^{\bullet\bullet} U = \frac{1}{2} \int_{\mathcal{X}} \left( \frac{W(y, z)}{c_W(\cdot, \cdot)} U(\cdot, (x, z)) + \frac{W(x, z)}{c_W(\cdot, \cdot)} U(\cdot, (y, z)) \right) d\mu(z),$$

*and the graph-graphon pairwise aggregation by*

$$M_X^{\bullet\bullet} U = \frac{1}{2N} \sum_{i=1}^{N} \left( \frac{W(y, X_i)}{c_X(\cdot, \cdot)} U(\cdot, (x, X_i)) + \frac{W(x, X_i)}{c_X(\cdot, \cdot)} U(\cdot, (y, X_i)) \right),$$

*and the sample-graph pairwise aggregation by*

$$M_A^{\bullet\bullet} U = \frac{1}{2N} \sum_{i=1}^{N} \left( \frac{A(y, X_i)}{c_A(\cdot, \cdot)} U(\cdot, (x, X_i)) + \frac{A(x, X_i)}{c_A(\cdot, \cdot)} U(\cdot, (y, X_i)) \right),$$

*where we define $A(x, X_i) \sim Ber(W(x, X_i))$ as a random variable.*

Maskey et al. [46, Definition A.7] has defined for a vector $\mathbf{z} = (z_1, \ldots, z_F) \in \mathbb{R}^F$, we define as usual

$$\|\mathbf{z}\|_\infty = \max_{1 \le k \le F} |z_k|.$$

For every $x, x' \in \mathcal{X}$, we say a function $f : \chi \to \mathbb{R}^F$ is Lipschitz continuous if there exists a $L_f > 0$ such that for every $x, x' \in \chi$, we have

$$\|f(x) - f(x')\|_\infty \le L_f d(x, x').$$

Here if $\chi = \mathbb{R}^F$, $d(x, x') = \|x - x'\|_\infty$. For our theoretical results, we make the following assumptions:

**Assumption 1.** *(extension of Maskey et al. [46, Definition A.10]) Let $(\chi, d)$ be a metric space and $W : \chi \times \chi \to [0, \infty)$. Let $\Theta$ be a MPNN with message and update functions $\Phi^{(l)} : \mathbb{R}^{2F_l} \to \mathbb{R}^{H_l}$ and $\Psi^{(l)} : \mathbb{R}^{F_l + H_l} \to \mathbb{R}^{F_{l+1}}$, $l = 1, \ldots, T - 1$.*

1. *By Definition 1 of the graphon , the graphon satisfies $\|W\|_\infty \le 1$.*

2. *[46, Definition A.10, item 6]: There exists a constant $\mathrm{d}_{\min} > 0$ such that for every $x \in \chi$, we have $d_W(x) \ge \mathrm{d}_{\min}$.*

3. *There exists a constant $d_{cmin}$ such that for every $x, y \in \mathcal{X}$, we have $c_W(x, y) \ge d_{cmin}$.*

4. *$M_{tr} = \max(supp(N^{tr}))$ is the largest graph in training, where $N^{tr}$ is the distribution of graph sizes in the training data.*

5. *[46, Similar to Definition A.10, item 7 adding dependence on $M_{tr}$]: For every $l = 1, \ldots, T$, the message function $\Phi^{(l)}$ and update function $\Psi^{(l)}$ are Lipschitz continuous with Lipschitz constants $L_\Phi^{(l)}(M_{tr})$ and $L_\Psi^{(l)}(M_{tr})$ respectively.*

## D  Large real-world and random graphs have relatively few isomorphic nodes

In what follows we show that isomorphic nodes are rare both in many real-world networks and SBMs. We start with real-world graphs. MacArthur et al. [42] has computed the fraction of *non-isomorphic* nodes (denoted as the *network redundancy* $r_\mathcal{G}$ by [42]) of different types of small (< 23,000 nodes) real-world graphs. MacArthur et al. [42] shows that the majority of biological graphs are composed of mostly non-isomorphic nodes. Small technological networks (e.g., road network) tend to have significantly more isomorphic nodes.

In order to see whether these results also hold for larger graphs, we performed a similar experiment on the following datasets.

- The ogbl-ppa dataset is an undirected, unweighted graph. Nodes represent proteins from 58 different species, and edges indicate biologically meaningful associations between proteins [74], e.g., physical interactions, co-expression, homology or genomic neighborhood.

- The ogbl-ddi dataset is a homogeneous, unweighted, undirected graph, representing the drug-drug interaction network. Each node represents an FDA-approved or experimental drug [74]. Edges represent interactions between drugs and can be interpreted as a phenomenon where the joint effect of taking the two drugs together is considerably different from the expected effect in which drugs act independently of each other.

- The Slashdot graph contains friend/foe links between the users of Slashdot (where we ignore edge types).

- HepPh is a co-authorship network where if an author $i$ co-authored a paper with author $j$, the graph contains a undirected edge from $i$ to $j$. If the paper is co-authored by $k$ authors this generates a completely connected (sub)graph on $k$ nodes.

- The Github graph shows GitHub developers (nodes) who have starred at least 10 repositories and edges are mutual follower relationships between them (we make the graph undirected for our analysis).

- The Twitch/En graph shows Twitch users (who stream in English) as nodes and links are mutual friendships between them.

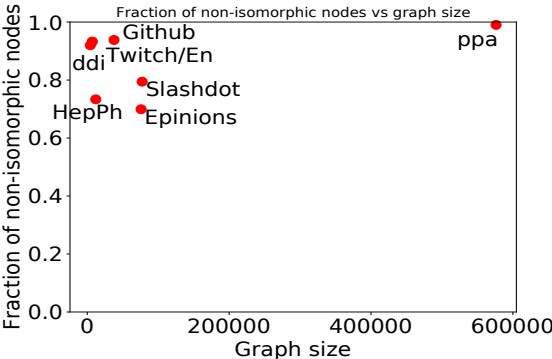

Figure 3: **Fraction of isomorphic nodes in real-world graphs:** The fraction of non-isomorphic nodes (also denoted as the *network redundancy* $r_G$ by [42]) in real-world graphs tends to be close to 90%, except in the HepPh collaboration network and Slashdot, which contain many small disconnected components. We assume all graphs are undirected and unattributed for this analysis.

- The Epinions graph shows users of the consumer review site Epinions.com as nodes and edges as trust relationships between users.

Figure 3 shows the fraction of *non-isomorphic* node shown against the size of the graph. This analysis considers a few datasets widely used in the neural network literature to benchmark link prediction methods such as OGB[2] ppa and ddi, where ppa is the largest dataset we were able to run the *nauty*[3] isomorphism checking algorithm without crashing. Nauty [48] is one of the most efficient graph isomorphism algorithms available, which we use to calculate a lower bound on the size of the automorphism group of our graphs. Using social networks in the SNAP[4] repository we again observe a large fraction of the nodes are non-isomorphic. Visual inspection shows that most isomorphic nodes are low-degree siblings (e.g., the most common are nodes with degree one that have the same parent). Note that our results do not contradict Ball and Geyer-Schulz [5], which shows that many real-world graphs have isomorphic nodes. Having isomorphic nodes is different than containing a large fraction of isomorphic nodes.

The results in Figure 3 show that most nodes in real-world graphs tend to be non-isomorphic for reasonably large graphs (in particular ppa). In what follows we show that random graph models generally *do not* contain isomorphic nodes with high probability.

*Theoretical results on random graphs.* Regarding isomorphic nodes on random graphs, we can prove the following result:

**Corollary 3.** *Consider a random graph $G = (V, E)$ with $N$ given nodes so that all possible $2^{\binom{N}{2}}$ graphs should have the same probability to be chosen. Then, as $N \to \infty$ all nodes in $G$ are non-isomorphic, regardless whether we take the nodes in $G$ to be attributed or unattributed.*

*Proof.* The proof for unattributed $G$ is a direct consequence of Erdos and Rényi [18, Theorem 2]. Adding node attributes cannot make two non-isomorphic nodes be isomorphic, which concludes our proof. □

Erdos and Rényi [18, Theorem 3] (see also Kim et al. [27, Theorem 3.1]) shows that the statement in Corollary 3 is also true for $G(N, p)$ graphs with p satisfying $(\ln N)/N \le p \le 1 - (\ln N)/N$. Kim et al. [27, Theorem 3.1] shows a similar result for random $d$-regular graph on $N$ vertices with $3 \le d \le n - 4$. Luczak et al. [41] has shown similar results for preferential-attachment graphs, where in each step a new node with $m \ge 3$ edges is added. In what follows we show a similar result for SBMs.

**Proposition 1.** *Consider a random graph $G = (V, E)$ with $N$ given nodes, generated by the SBM in Definition 6, where within-block and inter-block probabilities in $\boldsymbol{S}$ lie in the interval $(p, 1 - p)$, with*

---

[2]`https://ogb.stanford.edu/docs/graphprop/`

[3]https://pallini.di.uniroma1.it/

[4]https://snap.stanford.edu/data/index.html

$(\ln N)/N \leq p \leq 1 - (\ln N)/N$. *Then, as $N \to \infty$ all nodes in $G$ are non-isomorphic, regardless whether we take the nodes in $G$ to be attributed or unattributed.*

*Proof.* Let $\boldsymbol{S}$ have $r > 0$ blocks. Consider generating the $G$ first by sampling the within-block edges. Let $G_a$ be the induced subgraph of all nodes that belong to a single block $a \in \{1, \ldots, r\}$. By the results in Erdos and Rényi [18, Theorem 3], as $N \to \infty$, $G_a$ has no isomorphic nodes. The above is true for all within-block edges. Now consider sampling the between-block edges of two $i, j \in V$ nodes in $G$. The event of $i$ and $j$ being isomorphic (if considering just their edges to $G_a$) is the same as they connecting to the same nodes in $G_a$ (since each node on a block is non-isomorphic, if they connect to different nodes they would no longer be isomorphic). The probability of this event is at most $(1 - \epsilon)^{\alpha N}$, for $\epsilon = \min(p, 1 - p)$, where $\alpha > 0$ is the fraction of nodes in $G_a$ (which is not a function of $N$). As $N \to \infty$, by the union bound, the probability that this will happen with any pair of edges is at most $\binom{N}{2}(1 - \epsilon)^{\alpha N}$, which goes to zero.

W.l.o.g. now assume $a$ is the block with the least number of nodes (which is also diverging as $N \to \infty$). The only alternative for $i$ and $j$ to be isomorphic is to do so by connecting to nodes in distinct blocks. For instance, we could imagine $r$ copies of $G_a$: Even though $i$ and $j$ did not connect to the same nodes in the same graphs, they connected to their isomorphic equivalent nodes in different copies. But since there are only $r$ blocks, and $r$ does not depend on $N$, this event must have probability at most $(1 - \epsilon)^{\alpha N/r}$. As $N \to \infty$, by the union bound, the probability $\binom{N}{2}(1 - \epsilon)^{\alpha N/r}$ goes to zero. Replacing the copies of $G_a$ with the actual sampled blocks only makes this probability smaller, since the subgraphs of the other blocks are larger and may contain different topologies than $G_a$ (making their nodes distinct from the nodes in $G_a$). Finally, adding node attributes cannot make two non-isomorphic nodes be isomorphic, which concludes our proof. $\qquad\square$

# E    Proof results for Theorem 1

In what follows we provide the elements to prove Theorem 1.

First we prove the following lemma of the difference between a graph-graphon (N-normalized) sum aggregation and a sampled-graph (N-normalized) sum aggregation in Definition 13. Using the same assumptions as Maskey et al. [45, Lemma B.3]:

**Lemma 3.** *Let $(\chi, d, \mu)$ be a metric-measure space and $W$ be a graphon s.t. Assumptions 1.1-2. are satisfied. Let $\Phi : \mathbb{R}^{2F} \to \mathbb{R}^H$ be Lipschitz continuous with Lipschitz constant $L_\Phi(M_{tr})$, with $M_{tr}$ as in Assumption 1 item 4. Consider a metric-space signal $f : \chi \to \mathbb{R}^F$ with $\|f\|_\infty < \infty$. Suppose that $X_1, \ldots, X_N$ are drawn i.i.d. from $\mu$ on $\chi$, and let $p \in (0, 1/H)$. Let $x \in \chi$, and define the random variable*

$$T_x = \frac{1}{N} \sum_{i=1}^N A(x, X_i)\Phi\big(f(x), f(X_i)\big) - \frac{1}{N} \sum_{i=1}^N W(x, X_i)\Phi\big(f(x), f(X_i)\big)$$

*on the sample space $\mathcal{X}^N \times [0,1]^N$. Then, with probability at least $1 - Hp$, we have*

$$\|T_x\|_\infty \leq \sqrt{2}\frac{(L_\Phi(M_{tr})\|f\|_\infty + \|\Phi(0,0)\|_\infty)\sqrt{\log 2/p}}{\sqrt{N}}. \tag{5}$$

*Proof.* The proof of the bound is the same as the proof in Maskey et al. [45, Lemma B.3] since $\mathbb{E}(A(x, X_i)) = W(x, X_i)$, even though $T_x$ is a different quantity than the quantity used in Maskey et al. [45, Lemma B.3]. $\qquad\square$

Combining Maskey et al. [45, Lemma B.3] and Lemma 3, we can use the triangle inequality to prove the following lemma about concentration of error between a sampled-graph (N-normalized) sum aggregation in Definition 13 and the continuous integral aggregation in Definition 12, which is used in Definitions 3 and 5. Using the same assumption as Maskey et al. [45, Lemma B.4]:

**Lemma 4.** *Let $(\chi, d, \mu)$ be a metric-measure space and $W$ be a graphon s.t. Assumptions 1.1-2. are satisfied. Let $\Phi : \mathbb{R}^{2F} \to \mathbb{R}^H$ be Lipschitz continuous with Lipschitz constant $L_\Phi(M_{tr})$. Consider a*

metric-space signal $f : \chi \to \mathbb{R}^F$ with $\|f\|_\infty < \infty$. Suppose that $X_1, \ldots, X_N$ are drawn i.i.d. from $\mu$ on $\chi$, and let $p \in (0, 1/(2H))$. Let $x \in \chi$, and define the random variable

$$R_x = \frac{1}{N} \sum_{i=1}^N A(x, X_i) \Phi\big(f(x), f(X_i)\big) - \int_\chi W(x, y) \Phi\big(f(x), f(y)\big) d\mu(y)$$

on the sample space $\chi^N \times [0, 1]^N$. Then, with probability at least $1 - 2Hp$, we have

$$\|R_x\|_\infty \leq 2\sqrt{2} \frac{(L_\Phi(M_{tr})\|f\|_\infty + \|\Phi(0, 0)\|_\infty)\sqrt{\log 2/p}}{\sqrt{N}}. \tag{6}$$

*Proof.* Use the triangle inequality, the results from Maskey et al. [45, Lemma B.3] and Lemma 3. Define $Y_x = \frac{1}{N} \sum_{i=1}^N W(x, X_i) \Phi\big(f(x), f(X_i)\big) - \int_\chi W(x, y) \Phi\big(f(x), f(y)\big) d\mu(y)$.

$$\|R_x\|_\infty = \|T_x + Y_x\|_\infty \leq \|T_x\|_\infty + \|Y_x\|_\infty.$$

From Maskey et al. [45, Lemma B.3] and Lemma 3, $\|T_x\|_\infty \leq \sqrt{2} \frac{(L_\Phi(M_{tr})\|f\|_\infty + \|\Phi(0,0)\|_\infty)\sqrt{\log 2/p}}{\sqrt{N}}$
w.p. $1 - Hp$ and $\|Y_x\|_\infty \leq \sqrt{2} \frac{(L_\Phi(M_{tr})\|f\|_\infty + \|\Phi(0,0)\|_\infty)\sqrt{\log 2/p}}{\sqrt{N}}$ w.p. $1 - Hp$. We have with probability at least $1 - 2Hp$ using the union bound of the two events,

$$\|R_x\|_\infty \leq 2\sqrt{2} \frac{(L_\Phi(M_{tr})\|f\|_\infty + \|\Phi(0, 0)\|_\infty)\sqrt{\log 2/p}}{\sqrt{N}}.$$

$\square$

Based on Lemma 4, we can prove the following corollary about the maximum concentration error between sampled-graph (N-normalized) sum aggregation ($M_A^\bullet$) and continuous integral aggregation ($M_W^\bullet$) for all the nodes in the sampled graph $G$. Using the same overall framing as [45, Lemma B.3]:

**Corollary 4.** *Consider $(\chi, d, \mu)$ a metric-measure space and graphon $W$ satisfying items 1 and 2 of Assumption 1. Let $\Phi : \mathbb{R}^{2F} \to \mathbb{R}^H$ be Lipschitz continuous with Lipschitz constant $L_\Phi(M_{tr})$, and a metric-space signal $f : \chi \to \mathbb{R}^F$ with $\|f\|_\infty < \infty$. Define $X_1, \ldots, X_N$ as drawn i.i.d. from $\mu$ on $\chi$, and then edges $A_{i,j} \sim Ber(W(X_i, X_j))$ i.i.d sampled. Let $p \in (0, 1/2H)$, and define the random variable*

$$R_{X_i} = \frac{1}{N} \sum_{j=1}^N A(X_i, X_j) \Phi\big(f(X_i), f(X_j)\big) - \int_\chi W(X_i, y) \Phi\big(f(X_i), f(y)\big) d\mu(y)$$

*on the sample space $\chi^N \times [0, 1]^N$. Then, with probability at least $1 - 2Hp$, we have*

$$\max_{i=1,\ldots,N} \|(M_A^\bullet - M_W^\bullet)\big(\Phi(f, f)\big)(X_i)\|_\infty = \max_{i=1,\ldots,N} \|R_{X_i}\|_\infty$$

$$\leq 2\sqrt{2} \frac{(L_\Phi(M_{tr})\|f\|_\infty + \|\Phi(0, 0)\|_\infty)\sqrt{\log(2N/p)}}{\sqrt{N}}. \tag{7}$$

*Proof.* Using the result from Lemma 3 we have with probability $1 - \frac{Hp}{N}$,

$$\|\frac{1}{N} \sum_{i=1}^N A(x, X_i) \Phi\big(f(x), f(X_i)\big) - \frac{1}{N} \sum_{i=1}^N W(x, X_i) \Phi\big(f(x), f(X_i)\big)\|_\infty$$

$$\leq \sqrt{2} \frac{(L_\Phi(M_{tr})\|f\|_\infty + \|\Phi(0, 0)\|_\infty)\sqrt{\log(2N/p)}}{\sqrt{N}}.$$

Using the union bound of the $N$ events that the above equations happens for $x = X_1, \ldots, X_N$, with probability at least $1 - Hp$, we have

$$\max_{i=1,\ldots,N} \|\frac{1}{N} \sum_{j=1}^N A(X_i, X_j) \Phi\big(f(X_i), f(X_j)\big) - \frac{1}{N} \sum_{j=1}^N W(X_i, X_j) \Phi\big(f(X_i), f(X_j)\big)\|_\infty$$

$$\leq \sqrt{2} \frac{(L_\Phi(M_{tr})\|f\|_\infty + \|\Phi(0, 0)\|_\infty)\sqrt{\log(2N/p)}}{\sqrt{N}}.$$

The same logic can be applied to $Y_{X_i}, \forall i \in \{1, ..., N\}$. Thus, using the triangle inequality, and the union bound of the two events, we have with probability at least $1 - 2Hp$,

$$\max_{i=1,...,N} \|R_{X_i}\|_\infty \leq 2\sqrt{2} \frac{(L_\Phi(M_{\mathrm{tr}})\|f\|_\infty + \|\Phi(0,0)\|_\infty)\sqrt{\log(2N/p)}}{\sqrt{N}}.$$

$\square$

Now the layer-wise error between a cMPNN$^\bullet$ and gMPNN$^\bullet$ can be bounded as follows:

**Corollary 5.** *Consider $(\chi, d, \mu)$ a metric-measure space and graphon $W$ consistent with items 1 and 2 of Assumption 1. Let $\Phi : \mathbb{R}^{2F} \to \mathbb{R}^H$ and $\Psi : \mathbb{R}^{F+H} \to \mathbb{R}^{F'}$ be Lipschitz continuous with Lipschitz constants $L_\Phi(M_{tr})$ and $L_\Psi(M_{tr})$. Consider a metric-space signal $f : \chi \to \mathbb{R}^F$ with $\|f\|_\infty < \infty$. Let $p \in (0, \frac{1}{2(H+1)})$. Suppose that $X_1, \ldots, X_N$ are drawn i.i.d. from $\mu$ in $\chi$, and then edges $A_{i,j} \sim Ber(W(X_i, X_j))$ i.i.d sampled. Then with probability at least $1 - 2Hp$,*

$$\max_{i=1,...,N} \|\Psi\Big(f(\cdot), M_A^\bullet\big(\Phi(f,f)\big)(X_i)\Big) - \Psi\Big(f(\cdot), M_W^\bullet\big(\Phi(f,f)\big)(X_i)\Big)\|_\infty$$

$$\leq L_\Psi(M_{tr})\Big(2\sqrt{2} \frac{(L_\Phi(M_{tr})\|f\|_\infty + \|\Phi(0,0)\|_\infty)\sqrt{\log(2N/p)}}{\sqrt{N}}\Big). \tag{8}$$

*Proof.* The proof is the same as Maskey et al. [45, Lemma B.6]. The different result comes from the different bound between Corollary 4 and Maskey et al. [45, Lemma B.5]. $\square$

### E.1 Proof of Theorem 1

Following [45, Appendix B.2], they first bound the layer-wise error as Corollary 5, and derive the final bound through a recurrence relation. The only difference is on the layer-wise bound Corollary 6 and Maskey et al. [45, Corollary B.6]. We will omit the middle parts. Hence, finally, we can prove Theorem 1 by slightly adpating the proof in Maskey et al. [45, Theorem B.14] to our setting.

**Theorem 1** (OOD convergence without in-distribution convergence)**.** *For a random graph model $(W, f)$ satisfying Definition 1, let $N^{tr}$ be a random variable defining the distribution of graph sizes in training. Define the test distribution $(G^{te}, \boldsymbol{F}^{te}) \sim (W, f)$ through the causal graph in Figure 1 as an interventional change to obtain larger test graph sizes where $\min(supp(N^{te})) \gg M_{tr} = \max(supp(N^{tr}))$ (which means any test graph is much larger than the largest possible training graph). Let $\Theta = ((\Phi^{(l)})_{l=1}^T, (\Psi^{(l)})_{l=1}^T)$ be a MPNN as in Definition 2 with $T$ layers such that $\Phi^{(l)} : \mathbb{R}^{2F_{l-1}} \to \mathbb{R}^{H_{l-1}}$ and $\Psi^{(l)} : \mathbb{R}^{F_{l-1}+H_{l-1}} \to \mathbb{R}^{F_l}$ are learned from the training distribution and are Lipschitz continuous with Lipschitz constants $L_\Phi^{(l)}(M_{tr})$ and $L_\Psi^{(l)}(M_{tr})$ that depend on $M_{tr}$. Let gMPNN$^\bullet$ $\Theta_A^{\bullet(T)}$ and cMPNN$^\bullet$ $\Theta_W^{\bullet(T)}$ be as in Definitions 3 and 5. Let $X_1^{te}, ..., X_{N^{te}}^{te}$ and $\boldsymbol{A}^{te}$ be as in Definition 1. Let $p \in (0, \frac{1}{\sum_{l=1}^T 2(H_l+1)})$. Then, if*

$$\frac{\sqrt{N^{te}}}{\sqrt{\log(2N^{te}/p)}} \geq \frac{4\sqrt{2}}{d_{min}}, \tag{1}$$

*we have with probability at least $1 - \sum_{l=1}^T 2(H_l+1)p$,*

$$\delta_{A\text{-}W}^\bullet := \max_{i=1,...,N^{te}} \|\Theta_{\boldsymbol{A}^{te}}^{\bullet(T)}(\boldsymbol{F}^{te})_i - \Theta_W^{\bullet(T)}(f)(X_i^{te})\|_\infty \leq (C_1 + C_2\|f\|_\infty)\frac{\sqrt{\log(2N^{te}/p)}}{\sqrt{N^{te}}},$$

*where the constants $C_1$ and $C_2$ are defined in the Appendix and depend on $\{L_\Phi^{(l)}(M_{tr}), L_\Psi^{(l)}(M_{tr})\}_{l=1}^T$ and the distribution of $(G^{tr}, \boldsymbol{F}^{tr})$.*

*Proof.* In this case, $\|\Phi^{(l)}(0,0)\|_\infty, \|\Psi^{(l)}(0,0)\|_\infty$ can be determined by $(G^{tr}, \boldsymbol{F}^{tr})$, $N^{tr}$ if the MPNN $\Theta$ has been trained on the training graph $(G^{tr}, \boldsymbol{F}^{tr})$.

Following the procedure of Maskey et al. [45, Appendix B.2] with Corollary 5, we can derive similarly, with probability at least $1 - \sum_{l=1}^{T}(2H_l + 1)p$,

$$
\delta_{\text{A-W}}^\bullet \le \sum_{l=1}^{T} L_\Psi^{(l)}(M_{\text{tr}}) \left( 2\sqrt{2} \frac{(L_\Phi(M_{\text{tr}})^{(l)} \|f^{\bullet(l)}\|_\infty + \|\Phi^{(l)}(0,0)\|_\infty)\sqrt{\log(2N^{\text{te}}/p)}}{\sqrt{N^{\text{te}}}} \right)
$$
$$
\prod_{l'=l+1}^{T} ((L_\Psi^{(l')}(M_{\text{tr}}))^2 + 2(L_\Phi^{(l')}(M_{\text{tr}}))^2 (L_\Psi^{(l')}(M_{\text{tr}}))^2),
\tag{9}
$$

Using the same proof in Maskey et al. [45, Lemma B.9], we can derive

$$
\|f^{\bullet(l)}\|_\infty \le B_1^{(l)} + B_2^{(l)} \|f\|_\infty,
$$

where $B_1^{(l)}$, $B_2^{(l)}$ are independent of $f$, and

$$
B_1^{(l)} = \sum_{k=1}^{l} \left( L_\Psi^{(k)}(M_{\text{tr}}) \|\Phi^{(k)}(0,0)\|_\infty + \|\Psi^{(k)}(0,0)\|_\infty \right) \prod_{l'=k+1}^{l} L_\Psi^{(l')}(M_{\text{tr}}) \left( 1 + L_\Phi^{(l')}(M_{\text{tr}}) \right) \tag{10}
$$

and

$$
B_2^{(l)} = \prod_{k=1}^{l} L_\Psi^{(k)}(M_{\text{tr}}) \left( 1 + L_\Phi^{(k)}(M_{\text{tr}}) \right). \tag{11}
$$

Now we can decompose the summation in Equation (9). First, we defince $C_1$ as

$$
C_1 = \sum_{l=1}^{T} L_\Psi^{(l)}(M_{\text{tr}}) \left( 2\sqrt{2}(L_\Phi^{(l)}(M_{\text{tr}}) \, B_1^{(l)} + \|\Phi^{(l+1)}(0,0)\|_\infty) \right)
$$
$$
\times \prod_{l'=l+1}^{T} ((L_\Psi^{(l')}(M_{\text{tr}}))^2 + 2(L_\Phi^{(l')}(M_{\text{tr}}))^2 (L_\Psi^{(l')}(M_{\text{tr}}))^2),
\tag{12}
$$

Then we can define $C_2$ as

$$
C_2 = \sum_{l=1}^{T} L_\Psi^{(l)}(M_{\text{tr}}) \left( 2\sqrt{2} L_\Phi^{(l)}(M_{\text{tr}}) B_2^{(l)} \right) \prod_{l'=l+1}^{T} ((L_\Psi^{(l')}(M_{\text{tr}}))^2 + 2(L_\Phi^{(l')}(M_{\text{tr}}))^2 (L_\Psi^{(l')}(M_{\text{tr}}))^2),
\tag{13}
$$

It is clear to see we can rewrite Equation (9) as

$$
\delta_{\text{A-W}}^\bullet \le (C_1 + C_2 \|f\|_\infty) \frac{\sqrt{\log(2N^{\text{te}}/p)}}{\sqrt{N^{\text{te}}}}.
\tag{14}
$$

Thus $C_1$ and $C_2$ depends on $\{L_\Phi^{(l)}(M_{\text{tr}})\}_{l=1}^T$ and $\{L_\Psi^{(l)}(M_{\text{tr}})\}_{l=1}^T$. $\qquad\square$

# F   Proof of theoretical results for hardness of link prediction

In this section, we prove the results for $\Theta_A^{\bullet(T)}$ and $\Theta_W^{\bullet(T)}$ based on Theorem 1. Now we can prove Corollary 1.

**Corollary 1.** *Let* $\Theta = ((\Phi^{(l)})_{l=1}^T, (\Psi^{(l)})_{l=1}^T), \Theta_A^{\bullet(T)}, \Theta_W^{\bullet(T)}, p, (W, f), (G^{tr}, \boldsymbol{F}^{tr}), (G^{te}, \boldsymbol{F}^{te}), N^{tr},$ $N^{te}, A^{te},$ *and* $X_1^{te}, ..., X_{N^{te}}^{te}$ *be as in Theorem 1. If there exists* $i, j \in V^{te}, i \ne j$, *s.t.* $\Theta_W^{\bullet(T)}(X_i) = \Theta_W^{\bullet(T)}(X_j)$ *and Equation* (1) *is satisfied, then, with $C_1$ and $C_2$ as in Theorem 1, we have that with probability at least* $1 - \sum_{l=1}^{T} 2(H_l + 1)p$,

$$
\|\Theta_{\boldsymbol{A}^{te}}^{\bullet(T)}(\boldsymbol{F}^{te})_i - \Theta_{\boldsymbol{A}^{te}}^{\bullet(T)}(\boldsymbol{F}^{te})_j\|_\infty \le (C_1 + C_2 \|f\|_\infty) \frac{2\sqrt{\log(2N^{te}/p)}}{\sqrt{N^{te}}}.
$$

*Proof.* The proof follows Theorem 1 by using the triangle inequality.

From Theorem 1, we know with probability at least $1 - 2\sum_{l=1}^{T}(H_l + 1)p$, $\|\Theta_{A^{\text{te}}}^{\bullet(T)}(\boldsymbol{F}^{\text{te}})_i - \Theta_W^{\bullet(T)}(f)(X_i^{\text{te}})\|_\infty \leq (C_1 + C_2\|f\|_\infty)\frac{\sqrt{\log 2N^{\text{te}}/p}}{\sqrt{N^{\text{te}}}}$ and $\|\Theta_{A^{\text{te}}}^{\bullet(T)}(\boldsymbol{F}^{\text{te}})_j - \Theta_W^{\bullet(T)}(f)(X_j^{\text{te}})\|_\infty \leq (C_1 + C_2\|f\|_\infty)\frac{\sqrt{\log 2N^{\text{te}}/p}}{\sqrt{N^{\text{te}}}}$.

Then

$$
\begin{aligned}
&\|\Theta_{A^{\text{te}}}^{\bullet(T)}(\boldsymbol{F}^{\text{te}})_i - \Theta_{A^{\text{te}}}^{\bullet(T)}(\boldsymbol{F}^{\text{te}})_j\|_\infty \\
&\leq \|\Theta_{A^{\text{te}}}^{\bullet(T)}(\boldsymbol{F}^{\text{te}})_i - \Theta_W^{\bullet(T)}(f)(X_i^{\text{te}})\|_\infty + \|\Theta_W^{\bullet(T)}(f)(X_i^{\text{te}}) - \Theta_{A^{\text{te}}}^{\bullet(T)}(\boldsymbol{F}^{\text{te}})_j\|_\infty \\
&= \|\Theta_{A^{\text{te}}}^{\bullet(T)}(\boldsymbol{F}^{\text{te}})_i - \Theta_W^{\bullet(T)}(f)(X_i^{\text{te}})\|_\infty + \|\Theta_W^{\bullet(T)}(f)(X_j^{\text{te}}) - \Theta_{A^{\text{te}}}^{\bullet(T)}(\boldsymbol{F}^{\text{te}})_j\|_\infty \\
&\leq (C_1 + C_2\|f\|_\infty)\frac{2\sqrt{\log(2N^{\text{te}}/p)}}{\sqrt{N^{\text{te}}}}.
\end{aligned}
$$

The first inequality holds by traingle inequality, and the second equation holds since $\Theta_W^{\bullet(T)}(f)(X_i^{\text{te}}) = \Theta_W^{\bullet(T)}(f)(X_j^{\text{te}})$. $\qquad\square$

Then we are able to prove Lemma 1 by induction. By our Definition 7, we can also claim $t_k - t_{k-1} = t_{\pi(k)} - t_{\pi(k)-1}, \forall k \in \{1, ..., r\}$.

**Lemma 1.** *Let $\Theta = ((\Phi^{(l)})_{l=1}^T, (\Psi^{(l)})_{l=1}^T)$ be a MPNN as in Definition 2, and $\Theta_W^{\bullet(T)}$ as in Definition 5. For the SBM model $(W, f)$ in Definition 6 with $N^{\text{te}}$ nodes $X_1, \ldots, X_{N^{te}}$. If there exists $i, j \in V^{te}$ such that $X_i^{te}, X_j^{te}$ are nodes that belong to isomorphic SBM blocks (Definition 7), then $\Theta_W^{\bullet(T)}(f)(X_i^{te}) = \Theta_W^{\bullet(T)}(f)(X_j^{te})$.*

*Proof.* We prove the lemma by induction.

We assume in layer $l$, $f^{\bullet(l)}(X_i^{\text{te}}) = f^{\bullet(l)}(X_j^{\text{te}}), 1 \leq l \leq T - 1$, $f^{\bullet(l)}$ outputs the same value within each block $\boldsymbol{B}^{(l)}$, and $\boldsymbol{B}^{(l)} = \pi \circ \boldsymbol{B}^{(l)}$. By Definitions 6 and 7, we know the assumption holds for $l = 1$. First,

$$
f^{\bullet(l+1)}(X_i^{\text{te}}) = \Psi^{(l+1)}\Big(f^{\bullet(l)}(X_i^{\text{te}}), M_W^\bullet\big(\Phi^{(l+1)}(f^{\bullet(l)}, f^{\bullet(l)})\big)(X_i^{\text{te}})\Big).
$$

Since $f^{\bullet(l)}(X_i^{\text{te}}) = f^{\bullet(l)}(X_j^{\text{te}})$, we only need to show $M_W^\bullet\big(\Phi^{(l+1)}(f^{\bullet(l)}, f^{\bullet(l)})\big)(X_i^{\text{te}}) = M_W^\bullet\big(\Phi^{(l+1)}(f^{\bullet(l)}, f^{\bullet(l)})\big)(X_j^{\text{te}})$.

$$
\begin{aligned}
&M_W^\bullet\big(\Phi^{(l+1)}(f^{\bullet(l)}, f^{\bullet(l)})\big)(X_i^{\text{te}}) \\
&= \int_{[0,1]} W(X_i^{\text{te}}, y)\Phi^{(l+1)}(f^{\bullet(l)}(X_i^{\text{te}}), f^{\bullet(l)}(y))dy \\
&= \sum_{k=1}^r \int_{[t_{k-1}, t_k)} W(X_i^{\text{te}}, y)\Phi^{(l+1)}(f^{\bullet(l)}(X_i^{\text{te}}), f^{\bullet(l)}(y))dy \\
&= \sum_{k=1}^r \Phi^{(l+1)}(\boldsymbol{B}_a^{(l)}, \boldsymbol{B}_k^{(l)}) \int_{[t_{k-1}, t_k)} W(X_i^{\text{te}}, y)dy \\
&= \sum_{k=1}^r \Phi^{(l+1)}(\boldsymbol{B}_a^{(l)}, \boldsymbol{B}_k^{(l)})(t_k - t_{k-1})\boldsymbol{S}_{i,k} \\
&= \sum_{k=1}^r \Phi^{(l+1)}(\boldsymbol{B}_a^{(l)}, \boldsymbol{B}_k^{(l)})(t_k - t_{k-1})\boldsymbol{S}_{\pi(i),\pi(k)}
\end{aligned}
\tag{15}
$$

$$= \sum_{k=1}^{r} \Phi^{(l+1)}(\boldsymbol{B}_{\pi(a)}^{(l)}, \boldsymbol{B}_{\pi(k)}^{(l)})(t_{\pi(k)} - t_{\pi(k)-1})\boldsymbol{S}_{\pi(i),\pi(k)}$$

$$= \sum_{k=1}^{r} \Phi^{(l+1)}(\boldsymbol{B}_{b}^{(l)}, \boldsymbol{B}_{\pi(k)}^{(l)})(t_{\pi(k)} - t_{\pi(k)-1})\boldsymbol{S}_{j,\pi(k)}$$

$$= \sum_{k=1}^{r} \Phi^{(l+1)}(\boldsymbol{B}_{b}^{(l)}, \boldsymbol{B}_{k}^{(l)})(t_{k} - t_{k-1})\boldsymbol{S}_{j,k} \qquad (16)$$

$$= \sum_{k=1}^{r} \Phi^{(l+1)}(\boldsymbol{B}_{b}^{(l)}, \boldsymbol{B}_{k}^{(l)}) \int_{[t_{k-1},t_k)} W(X_j^{\text{te}}, y)dy$$

$$= \int_{[0,1]} W(X_j^{\text{te}}, y)\Phi^{(l+1)}(f^{\bullet(l)}(X_j^{\text{te}}), f^{\bullet(l)}(y))dy$$

$$= M_W^{\bullet}\big(\Phi^{(l+1)}(f^{\bullet(l)}, f^{\bullet(l)})\big)(X_j^{\text{te}})$$

Here we use the fact that $f^{\bullet(l)}$ $f^{\bullet(l)}$ outputs the same value within each block, and $\boldsymbol{B}_{k}^{(l)} = \boldsymbol{B}_{\pi(k)}^{(l)}, \forall k \in \{1, ..., r\}$.

We have shown $f^{\bullet(l+1)}(X_i^{\text{te}}) = f^{\bullet(l+1)}(X_j^{\text{te}})$. And this proof applies for all $X_i^{\text{te}} \in [t_{a-1}, t_a)$ (in block $a$), and the same conclusion holds. So $f^{\bullet(l+1)}$ still outputs the same value within each block. Furthermore, $\boldsymbol{B}_{a}^{(l+1)} = \boldsymbol{B}_{\pi(a)}^{(l+1)}$ using the same proof technique. And this implies $\pi \circ \boldsymbol{B}^{(l+1)} = \boldsymbol{B}^{(l+1)}$.

Thus, $\Theta_W^{\bullet(T)}(X_i^{\text{te}}) = f^{\bullet(T)}(X_i^{\text{te}}) = f^{\bullet(T)}(X_j^{\text{te}}) = \Theta_W^{\bullet(T)}(X_j^{\text{te}})$. □

Then we are ready to prove Corollary 2 by applying Corollary 1.

**Corollary 2.** *Let $\Theta = ((\Phi^{(l)})_{l=1}^{T}, (\Psi^{(l)})_{l=1}^{T})$ be the MPNN with $T$ layers and $\Theta_A^{\bullet(T)}, \Theta_W^{\bullet(T)}$ as in Theorem 1. Let $\eta^\bullet : \mathbb{R}^{F_T} \times \mathbb{R}^{F_T} \to [0, 1]$ be as in Definition 8. Consider the SBM $(W, f)$ in Definition 6 with isomorphic blocks (Definition 7). Let $(G^{tr}, \boldsymbol{F}^{tr}) \sim (W, f)$ and $(G^{te}, \boldsymbol{F}^{te}) \sim (W, f)$ be the training and test graphs with $N^{tr}$ and $N^{te}$ nodes, respectively as in Theorem 1. Consider any two test nodes $i, j \in \{1, ..., N^{te}\}$, $i \neq j$, for which we can make a link prediction decision with $\eta^\bullet$ (i.e., $\eta^\bullet(\Theta_{A^{te}}^{\bullet(T)}(\boldsymbol{F}^{te})_i, \Theta_A^{\bullet(T)}(\boldsymbol{F}^{te})_j) \neq \tau$). Let $G^{te}$ be large enough to satisfy both Equation (1) and*

$$\frac{\sqrt{N^{te}}}{\sqrt{\log(2N^{te}/p)}} > \frac{2(C_1 + C_2\|f\|_\infty)}{|\eta^\bullet(\Theta_{A^{te}}^{\bullet(T)}(\boldsymbol{F}^{te})_i, \Theta_{A^{te}}^{\bullet(T)}(\boldsymbol{F}^{te})_j) - \tau|/L_\eta^\bullet(M_{tr})},$$

*where $p$, $C_1$, and $C_2$ are as given in Corollary 1. Then, if $i$ and $j$ belong to isomorphic blocks (i.e., $\Theta_W^{\bullet(T)}(f)(X_i^{te}) = \Theta_W^{\bullet(T)}(f)(X_j^{te})$), with probability at least $1 - \sum_{l=1}^{T} 2(H_l + 1)p$ the link prediction method in Definition 8 will make the same link prediction regardless of the SBM probability matrix $\boldsymbol{S}$ (Definition 6) and whether $i$ and $j$ are in the same block or distinct isomorphic blocks.*

*Proof.* To prove the corollary, we assume we have two nodes $j$ and $j'$, such that $i$ and $j$ are in the same block while $i$ and $j'$ are in distinct isomorphic blocks. In the proof, we will show that the link prediction between $i$ and $j$ and the prediction between $i$ and $j'$ will be the same.

First, from Corollary 1, since nodes $j$ and $j'$ are in distinct isomorphic SBM blocks, when Equation (1) is satisfied, we have with probability at least $1 - 2\sum_{l=1}^{T}(H_l + 1)p$

$$\|\Theta_{A^{te}}^{\bullet(T)}(\boldsymbol{F}^{te})_j - \Theta_{A^{te}}^{\bullet(T)}(\boldsymbol{F}^{te})_{j'}\|_\infty \leq (C_1 + C_2\|f\|_\infty)\frac{2\sqrt{\log 2N^{te}/p}}{\sqrt{N^{te}}}.$$

Then when the requirement for $N^{\text{te}}$ is satisfied,

$$\|\eta^{\bullet}(\Theta_{A^{\text{te}}}^{\bullet(T)}(\boldsymbol{F}^{\text{te}})_i, \Theta_{A^{\text{te}}}^{\bullet(T)}(\boldsymbol{F}^{\text{te}})_j) - \eta^{\bullet}(\Theta_{A^{\text{te}}}^{\bullet(T)}(\boldsymbol{F}^{\text{te}})_i, \Theta_{A^{\text{te}}}^{\bullet(T)}(\boldsymbol{F}^{\text{te}})_{j'})\|_{\infty}$$

$$\leq L_{\eta}^{\bullet}(M_{\text{tr}})\|\Theta_{A^{\text{te}}}^{\bullet(T)}(\boldsymbol{F}^{\text{te}})_j - \Theta_{A^{\text{te}}}^{\bullet(T)}(\boldsymbol{F}^{\text{te}})_{j'}\|_{\infty} \leq L_{\eta}^{\bullet}(M_{\text{tr}})(C_1 + C_2\|f\|_{\infty})\frac{2\sqrt{\log 2N^{\text{te}}/p}}{\sqrt{N^{\text{te}}}}$$

$$< \|\eta^{\bullet}(\Theta_{A^{\text{te}}}^{\bullet(T)}(\boldsymbol{F}^{\text{te}})_i, \Theta_{A^{\text{te}}}^{\bullet(T)}(\boldsymbol{F}^{\text{te}})_j) - \tau\|_{\infty}$$

If $\eta^{\bullet}(\Theta_{A^{\text{te}}}^{\bullet(T)}(\boldsymbol{F}^{\text{te}})_i, \Theta_{A^{\text{te}}}^{\bullet(T)}(\boldsymbol{F}^{\text{te}})_j) > \tau$, then

$$\eta^{\bullet}(\Theta_{A^{\text{te}}}^{\bullet(T)}(\boldsymbol{F}^{\text{te}})_i, \Theta_{A^{\text{te}}}^{\bullet(T)}(\boldsymbol{F}^{\text{te}})_{j'})$$

$$\geq \eta^{\bullet}(\Theta_{A^{\text{te}}}^{\bullet(T)}(\boldsymbol{F}^{\text{te}})_i, \Theta_{A^{\text{te}}}^{\bullet(T)}(\boldsymbol{F}^{\text{te}})_j) - |\eta^{\bullet}(\Theta_{A^{\text{te}}}^{\bullet(T)}(\boldsymbol{F}^{\text{te}})_i, \Theta_{A^{\text{te}}}^{\bullet(T)}(\boldsymbol{F}^{\text{te}})_j) - \eta^{\bullet}(\Theta_{A^{\text{te}}}^{\bullet(T)}(\boldsymbol{F}^{\text{te}})_i, \Theta_{A^{\text{te}}}^{\bullet(T)}(\boldsymbol{F}^{\text{te}})_{j'})|$$

$$> \eta^{\bullet}(\Theta_{A^{\text{te}}}^{\bullet(T)}(\boldsymbol{F}^{\text{te}})_i, \Theta_{A^{\text{te}}}^{\bullet(T)}(\boldsymbol{F}^{\text{te}})_j) - |\eta^{\bullet}(\Theta_{A^{\text{te}}}^{\bullet(T)}(\boldsymbol{F}^{\text{te}})_i, \Theta_{A^{\text{te}}}^{\bullet(T)}(\boldsymbol{F}^{\text{te}})_j) - \tau|$$

$$= \eta^{\bullet}(\Theta_{A^{\text{te}}}^{\bullet(T)}(\boldsymbol{F}^{\text{te}})_i, \Theta_{A^{\text{te}}}^{\bullet(T)}(\boldsymbol{F}^{\text{te}})_j) - (\eta^{\bullet}(\Theta_{A^{\text{te}}}^{\bullet(T)}(\boldsymbol{F}^{\text{te}})_i, \Theta_{A^{\text{te}}}^{\bullet(T)}(\boldsymbol{F}^{\text{te}})_j) - \tau) = \tau$$

If $\eta^{\bullet}(\Theta_{A^{\text{te}}}^{\bullet(T)}(\boldsymbol{F}^{\text{te}})_i, \Theta_{A^{\text{te}}}^{\bullet(T)}(\boldsymbol{F}^{\text{te}})_j) < \tau$, then

$$\eta^{\bullet}(\Theta_{A^{\text{te}}}^{\bullet(T)}(\boldsymbol{F}^{\text{te}})_i, \Theta_{A^{\text{te}}}^{\bullet(T)}(\boldsymbol{F}^{\text{te}})_{j'})$$

$$\leq \eta^{\bullet}(\Theta_{A^{\text{te}}}^{\bullet(T)}(\boldsymbol{F}^{\text{te}})_i, \Theta_{A^{\text{te}}}^{\bullet(T)}(\boldsymbol{F}^{\text{te}})_j) + |\eta^{\bullet}(\Theta_{A^{\text{te}}}^{\bullet(T)}(\boldsymbol{F}^{\text{te}})_i, \Theta_{A^{\text{te}}}^{\bullet(T)}(\boldsymbol{F}^{\text{te}})_j) - \eta^{\bullet}(\Theta_{A^{\text{te}}}^{\bullet(T)}(\boldsymbol{F}^{\text{te}})_i, \Theta_{A^{\text{te}}}^{\bullet(T)}(\boldsymbol{F}^{\text{te}})_{j'})|$$

$$< \eta^{\bullet}(\Theta_{A^{\text{te}}}^{\bullet(T)}(\boldsymbol{F}^{\text{te}})_i, \Theta_{A^{\text{te}}}^{\bullet(T)}(\boldsymbol{F}^{\text{te}})_j) + |\eta^{\bullet}(\Theta_{A^{\text{te}}}^{\bullet(T)}(\boldsymbol{F}^{\text{te}})_i, \Theta_{A^{\text{te}}}^{\bullet(T)}(\boldsymbol{F}^{\text{te}})_j) - \tau|$$

$$= \eta^{\bullet}(\Theta_{A^{\text{te}}}^{\bullet(T)}(\boldsymbol{F}^{\text{te}})_i, \Theta_{A^{\text{te}}}^{\bullet(T)}(\boldsymbol{F}^{\text{te}})_j) - (\eta^{\bullet}(\Theta_{A^{\text{te}}}^{\bullet(T)}(\boldsymbol{F}^{\text{te}})_i, \Theta_{A^{\text{te}}}^{\bullet(T)}(\boldsymbol{F}^{\text{te}})_j) - \tau) = \tau$$

So whether $i$ and $j$ are in the same block, or in distinct isomorphic SBM blocks, their prediction will be the same (both links have predictions larger than $\tau$ or less). $\qquad\square$

## G Proof for pairwise gMPNN$^{\bullet\bullet}$ and cMPNN$^{\bullet\bullet}$

First we prove Lemma 2 showing $W(x, y)$ is a stationary point in cMPNN$^{\bullet\bullet}$.

**Lemma 2.** *If $\Phi(x, y) = y$ and $\Psi(x, y) = x/y$, then $f^{\bullet\bullet(t)}(x, y) = W(x, y), \forall x, y \in \mathcal{X}$ is a stationary point in the cMPNN$^{\bullet\bullet}$, i.e. if $f^{\bullet\bullet(t-1)}(x, y) = W(x, y)$, then $f^{\bullet\bullet(t)}(x, y) = W(x, y), \forall x, y \in \mathcal{X}$.*

*Proof.* If $f^{\bullet\bullet(t-1)}(x, y) = W(x, y)$, then

$$M_W^{\bullet\bullet}(\Phi^{(t)}(f^{(t-1)}))(x, y) = \frac{1}{2}\int_{\mathcal{X}}\left(\frac{W(y, z)}{c_W(x, y)}\Phi^{(t)}(f^{\bullet\bullet(t-1)}(x, y), f^{\bullet\bullet(t-1)}(x, z))\right.$$

$$\left. + \frac{W(x, z)}{c_W(x, y)}\Phi^{(t)}(f^{\bullet\bullet(t-1)}(x, y), f^{\bullet\bullet(t-1)}(y, z))\right)d\mu(z)$$

$$= \frac{1}{2}\int_{\mathcal{X}}\left(\frac{W(y, z)}{c_W(x, y)}W(x, z) + \frac{W(x, z)}{c_W(x, y)}W(y, z)\right)d\mu(z)$$

$$= \frac{1}{c_W(x, y)}\int_{\mathcal{X}}W(x, z)W(y, z)d\mu(z)$$

$$= \frac{c_W(x, y)}{c_W(x, y)} = 1$$

Thus $f^{\bullet\bullet(t)}(x, y) = \Psi^{(t)}(f^{\bullet\bullet(t-1)}(x, y), M_W^{\bullet\bullet}(\Phi^{(t)}(f^{\bullet\bullet(t-1)}, f^{\bullet\bullet(t-1)}))(x, y)) = \frac{W(x, y)}{1} = W(x, y)$.

We finish proving $W(x, y)$ is a staninoary point in cMPNN$^{\bullet\bullet}$. There are infinity choices of $\Phi$ and $\Psi$ such that $W(x, y)$ is a staninoary point. $\qquad\square$

Then we aim to prove Theorem 2, and the prove procedure should be very similar as Theorem 1.

## G.1 Preparation

Following Maskey et al. [45, Lemma B.3], we propose the following lemma for cMPNN$^{\bullet\bullet}$. Using the same overall framing as Maskey et al. [45, Lemma B.3],

**Lemma 5.** *Let $(\chi, d, \mu)$ be a metric-measure space and $W$ be a graphon s.t. Assumptions 1.1-3. are satisfied. Let $\Phi : \mathbb{R}^{2F} \to \mathbb{R}^H$ be Lipschitz continuous with Lipschitz constant $L_\Phi(M_{tr})$. Consider a metric-space signal $f^{\bullet\bullet} : \chi \times \chi \to \mathbb{R}^F$ with $\|f^{\bullet\bullet}\|_\infty < \infty$. Suppose that $X_1, \ldots, X_N$ are drawn i.i.d. from $\mu$ on $\chi$, and let $p \in (0, 1/H)$. Let $x, y \in \chi$, and define the random variable*

$$Y_{x,y}^{\bullet\bullet} = \frac{1}{2N} \sum_{i=1}^N \Big( W(y, X_i) \Phi(f^{\bullet\bullet}(x, y), f^{\bullet\bullet}(x, X_i)) + W(x, X_i) \Phi(f^{\bullet\bullet}(x, y), f^{\bullet\bullet}(y, X_i)) \Big)$$

$$- \frac{1}{2} \int_\chi \Big( W(y, z) \Phi(f^{\bullet\bullet}(x, y), f^{\bullet\bullet}(x, z))) + W(x, z) \Phi(f^{\bullet\bullet}(x, y), f^{\bullet\bullet}(y, z)) \Big) d\mu(z)$$

*on the sample space $\chi^N$. Then, with probability at least $1 - Hp$, we have*

$$\|Y_{x,y}^{\bullet\bullet}\|_\infty \leq \sqrt{2} \frac{(L_\Phi(M_{tr}) \|f^{\bullet\bullet}\|_\infty + \|\Phi(0,0)\|_\infty) \sqrt{\log 2/p}}{\sqrt{N}}. \tag{17}$$

*Proof.* The proof of the bound is the same as the proof in Maskey et al. [45, Lemma B.3], even though $Y_{x,y}$ is a different quantity than the quantity used in Maskey et al. [45, Lemma B.3]. $\quad\square$

**Lemma 6.** *Let $(\chi, d, \mu)$ be a metric-measure space and $W$ be a graphon s.t. Assumptions 1.1-3. are satisfied. Let $\Phi : \mathbb{R}^{2F} \to \mathbb{R}^H$ be Lipschitz continuous with Lipschitz constant $L_\Phi(M_{tr})$. Consider a metric-space signal $f^{\bullet\bullet} : \chi \times \chi \to \mathbb{R}^F$ with $\|f^{\bullet\bullet}\|_\infty < \infty$. Suppose that $X_1, \ldots, X_N$ are drawn i.i.d. from $\mu$ on $\chi$, and let $p \in (0, 1/H)$. Let $x, y \in \chi$, and define the random variable*

$$T_{x,y}^{\bullet\bullet} = \frac{1}{2N} \sum_{i=1}^N \Big( A(y, X_i) \Phi(f^{\bullet\bullet}(x, y), f^{\bullet\bullet}(x, X_i)) + A(x, X_i) \Phi(f^{\bullet\bullet}(x, y), f^{\bullet\bullet}(y, X_i)) \Big)$$

$$- \frac{1}{2N} \sum_{i=1}^N \Big( W(y, X_i) \Phi(f^{\bullet\bullet}(x, y), f^{\bullet\bullet}(x, X_i)) + W(x, X_i) \Phi(f^{\bullet\bullet}(x, y), f^{\bullet\bullet}(y, X_i)) \Big)$$

*on the sample space $\mathcal{X}^N \times [0, 1]^{2N}$. Then, with probability at least $1 - Hp$, we have*

$$\|T_{x,y}^{\bullet\bullet}\|_\infty \leq \sqrt{2} \frac{(L_\Phi(M_{tr}) \|f^{\bullet\bullet}\|_\infty + \|\Phi(0,0)\|_\infty) \sqrt{\log 2/p}}{\sqrt{N}}. \tag{18}$$

*Proof.* The proof procedure is the same as Maskey et al. [45, Lemma B.3] where we use $\mathbb{E}[A(y, X_i)] = W(y, X_i)$ and $\mathbb{E}[A(x, X_i)] = W(x, X_i)$. $\quad\square$

**Lemma 7.** *Let $(\chi, d, \mu)$ be a metric-measure space and $W$ be a graphon s.t. Assumptions 1.1-3. are satisfied. Let $\Phi : \mathbb{R}^{2F} \to \mathbb{R}^H$ be Lipschitz continuous with Lipschitz constant $L_\Phi(M_{tr})$. Consider a metric-space signal $f^{\bullet\bullet} : \chi \times \chi \to \mathbb{R}^F$ with $\|f^{\bullet\bullet}\|_\infty < \infty$. Suppose that $X_1, \ldots, X_N$ are drawn i.i.d. from $\mu$ on $\chi$, and let $p \in (0, 1/(2H))$. Let $x, y \in \chi$, and define the random variable*

$$R_{x,y}^{\bullet\bullet} = \frac{1}{2N} \sum_{i=1}^N \Big( A(y, X_i) \Phi(f^{\bullet\bullet}(x, y), f^{\bullet\bullet}(x, X_i)) + A(x, X_i) \Phi(f^{\bullet\bullet}(x, y), f^{\bullet\bullet}(y, X_i)) \Big)$$

$$- \frac{1}{2} \int_\chi \Big( W(y, z) \Phi(f^{\bullet\bullet}(x, y), f^{\bullet\bullet}(x, z))) + W(x, z) \Phi(f^{\bullet\bullet}(x, y), f^{\bullet\bullet}(y, z)) \Big) d\mu(z)$$

*on the sample space $\chi^N \times [0, 1]^{2N}$. Then, with probability at least $1 - 2Hp$, we have*

$$\|R_{x,y}^{\bullet\bullet}\|_\infty \leq 2\sqrt{2} \frac{(L_\Phi(M_{tr}) \|f^{\bullet\bullet}\|_\infty + \|\Phi(0,0)\|_\infty) \sqrt{\log 2/p}}{\sqrt{N}}. \tag{19}$$

*Proof.* Use the triangle inequality and the results from Lemmas 5 and 6.

$$\|R_{x,y}^{\bullet\bullet}\|_\infty = \|T_{x,y}^{\bullet\bullet} + Y_{x,y}^{\bullet\bullet}\|_\infty \le \|T_{x,y}^{\bullet\bullet}\|_\infty + \|Y_{x,y}^{\bullet\bullet}\|_\infty.$$

From Lemmas 5 and 6, $\|T_{x,y}^{\bullet\bullet}\|_\infty \le \sqrt{2}\frac{(L_\Phi(M_{\mathrm{tr}})\|f^{\bullet\bullet}\|_\infty + \|\Phi(0,0)\|_\infty)\sqrt{\log 2/p}}{\sqrt{N}}$ w.p. $1 - Hp$ and $\|Y_{x,y}^{\bullet\bullet}\|_\infty \le \sqrt{2}\frac{(L_\Phi(M_{\mathrm{tr}})\|f^{\bullet\bullet}\|_\infty + \|\Phi(0,0)\|_\infty)\sqrt{\log 2/p}}{\sqrt{N}}$ w.p. $1 - Hp$. With probability at least $1 - 2Hp$, by intersecting the two events, we have

$$\|R_{x,y}^{\bullet\bullet}\|_\infty \le 2\sqrt{2}\frac{(L_\Phi(M_{\mathrm{tr}})\|f^{\bullet\bullet}\|_\infty + \|\Phi(0,0)\|_\infty)\sqrt{\log 2/p}}{\sqrt{N}}.$$

□

Based on Lemma 7, we can prove the following corollary about the maximum concentration error for all pairs of nodes in the sampled graph $G$. Using the same overall framing as Maskey et al. [45, Lemma B.3],

**Corollary 6.** *Let $(\chi, d, \mu)$ be a metric-measure space and $W$ be a graphon s.t. Assumptions 1.1-3. are satisfied. Let $\Phi : \mathbb{R}^{2F} \to \mathbb{R}^H$ be Lipschitz continuous with Lipschitz constant $L_\Phi(M_{tr})$. Consider a metric-space signal $f^{\bullet\bullet} : \chi \times \chi \to \mathbb{R}^F$ with $\|f^{\bullet\bullet}\|_\infty < \infty$. Suppose that $X_1, \ldots, X_N$ are drawn i.i.d. from $\mu$ on $\chi$, and then edges $A_{i,j} \sim Ber(W(X_i, X_j))$ i.i.d sampled. Let $p \in (0, 1/2H)$, and define the random variable*

$$
\begin{aligned}
R_{X_i, X_j}^{\bullet\bullet} = \frac{1}{2N} \sum_{z=1}^N & \Big( A(X_j, X_z)\Phi(f^{\bullet\bullet}(X_i, X_j), f^{\bullet\bullet}(X_i, X_z)) \\
& + A(X_i, X_z)\Phi(f^{\bullet\bullet}(X_i, X_j), f^{\bullet\bullet}(X_j, X_z)) \Big) \\
& - \frac{1}{2} \int_\chi \Big( W(X_j, z)\Phi(f^{\bullet\bullet}(X_i, X_j), f^{\bullet\bullet}(X_i, z))) \\
& + W(X_i, z)\Phi(f^{\bullet\bullet}(X_i, X_j), f^{\bullet\bullet}(X_j, z)) \Big) d\mu(z)
\end{aligned}
$$

*on the sample space $\chi^N \times [0,1]^{2N}$. Then, with probability at least $1 - 2Hp$, we have*

$$\max_{i,j=1,\ldots,N} \|R_{X_i, X_j}^{\bullet\bullet}\|_\infty \le 2\sqrt{2}\frac{(L_\Phi(M_{tr})\|f^{\bullet\bullet}\|_\infty + \|\Phi(0,0)\|_\infty)\sqrt{\log(2N^2/p)}}{\sqrt{N}}. \tag{20}$$

*Proof.* Using the result from Lemma 6, we have with probability $1 - \frac{Hp}{N^2}$,

$$\|T_{X_i, X_j}^{\bullet\bullet}\|_\infty \le \sqrt{2}\frac{(L_\Phi(M_{\mathrm{tr}})\|f^{\bullet\bullet}\|_\infty + \|\Phi(0,0)\|_\infty)\sqrt{\log(2N^2/p)}}{\sqrt{N}}.$$

Using the union bound of the $N^2$ events that the above equations happens for $x = X_1, \ldots, X_N$ and $y = X_1, \ldots, X_N$, with probability at least $1 - Hp$, we have

$$\max_{i,j=1,\ldots,N} \|T_{X_i, X_j}^{\bullet\bullet}\|_\infty \le \sqrt{2}\frac{(L_\Phi(M_{\mathrm{tr}})\|f^{\bullet\bullet}\|_\infty + \|\Phi(0,0)\|_\infty)\sqrt{\log(2N^2/p)}}{\sqrt{N}}.$$

The same logic can be applied to $Y_{X_i, X_j}^{\bullet\bullet}, \forall i \in \{1, \ldots, N\}$. Thus, using the triangle inequality, and the union bound of the two events, we have with probability at least $1 - 2Hp$,

$$\max_{i,j=1,\ldots,N} \|R_{X_i, X_j}^{\bullet\bullet}\|_\infty \le 2\sqrt{2}\frac{(L_\Phi(M_{\mathrm{tr}})\|f^{\bullet\bullet}\|_\infty + \|\Phi(0,0)\|_\infty)\sqrt{\log(2N^2/p)}}{\sqrt{N}}.$$

□

Following Maskey et al. [45, Lemma B.2], we also bound the maximum sampled-graph fraction of common neighbors $c_A(\cdot, \cdot)$ under a condition of the sample size $N$. Using a same assumption as Maskey et al. [45, Lemma B.2],

**Lemma 8.** *Let $(\chi, d, \mu)$ be a metric-measure space and $W$ be a graphon s.t. Assumptions 1.1-3. are satisfied. Suppose that $X_1, \ldots, X_N$ are drawn i.i.d. from $\mu$ on $\chi$, and then edges $A_{i,j} \sim Ber(W(X_i, X_j))$ i.i.d sampled. And let $p \in (0, 1)$. If $N \in \mathbb{N}$ satisfy*

$$\sqrt{N} \geq 4\sqrt{2}\frac{\sqrt{\log(2N^2/p)}}{d_{cmin}}. \tag{21}$$

*Then, with probability at least $1 - 2p$, we have*

$$\max_{i,j=1,\ldots,N} \|c_A(X_i, X_j) - c_W(X_i, X_j)\|_\infty \leq 2\sqrt{2}\frac{\sqrt{\log(2N^2/p)}}{\sqrt{N}},$$

*and*

$$\min_{i,j=1,\ldots,N} c_A(X_i, X_j) \geq \frac{d_{cmin}}{2}. \tag{22}$$

*Proof.* For given $x, y \in \mathcal{X}$, define the random variable

$$c_A(x, y) - c_X(x, y) = \frac{1}{N}\sum_{i=1}^{N} A(x, X_i)A(y, X_i) - \frac{1}{N}\sum_{i=1}^{N} W(x, X_i)W(y, X_i)$$

on the sample space $\chi^N \times [0, 1]^{2N}$. Using the same proof technique in Lemmas 5 to 7, we can prove with probability at least $1 - 2p$, we have

$$\max_{i,j=1,\ldots,N} \|c_A(X_i, X_j) - c_W(X_i, X_j)\|_\infty \leq 2\sqrt{2}\frac{\sqrt{\log(2N^2/p)}}{\sqrt{N}},$$

Since $c_W(X_i, X_j) \geq d_{\text{cmin}}$, then with probability at least $1 - 2p$, when Equation (21) is satisfied, we have

$$\min_{i,j=1,\ldots,N} c_A(X_i, X_j) \geq \frac{d_{\text{cmin}}}{2}$$

$\square$

Based on Lemma 8, we can prove a modified version of Maskey et al. [45, Lemma B.5]. Using a same overall framing as Maskey et al. [45, Lemma B.5],

**Lemma 9.** *Let $(\chi, d, \mu)$ be a metric-measure space and $W$ be a graphon s.t. Assumptions 1.1-3. are satisfied. Let $\Phi : \mathbb{R}^{2F} \to \mathbb{R}^H$ be Lipschitz continuous with Lipschitz constant $L_\Phi(M_{tr})$. Consider a metric-space signal $f^{\bullet\bullet} : \chi \times \chi \to \mathbb{R}^F$ with $\|f^{\bullet\bullet}\|_\infty < \infty$. Let $p \in (0, \frac{1}{2(H+1)})$, and let $N \in \mathbb{N}$ satisfy (21). Suppose that $X_1, \ldots, X_N$ are drawn i.i.d. from $\mu$ in $\chi$, and then edges $A_{i,j} \sim Ber(W(X_i, X_j))$ i.i.d sampled. Then, condition (22) together with (23) below are satisfied in probability at least $1 - 2(H + 1)p$:*

$$\max_{i,j=1,\ldots,N} \|(M_A^{\bullet\bullet} - M_W^{\bullet\bullet})(\Phi(f^{\bullet\bullet}, f^{\bullet\bullet}))(X_i, X_j)\|_\infty$$
$$\leq 4\frac{\sqrt{2}\sqrt{\log(2N^2/P)}}{\sqrt{N}d_{cmin}^2}(L_\Phi(M_{tr})\|f^{\bullet\bullet}\|_\infty + \|\Phi(0,0)\|_\infty) \tag{23}$$
$$+ \frac{2\sqrt{2}(L_\Phi(M_{tr})\|f^{\bullet\bullet}\|_\infty + \|\Phi(0,0)\|_\infty)\sqrt{\log(2N^2/P)}}{d_{cmin}\sqrt{N}},$$

*Proof.* The proof is the same as Maskey et al. [45, Lemma B.5]. The only difference is on the difference between Lemma 8 and Maskey et al. [45, Lemma B.2], and the difference between Corollary 6 and Maskey et al. [45, Lemma B.4]. $\square$

Same as Maskey et al. [45, Lemma B.6], the layer-wise error for cMPNN$^{\bullet\bullet}$ and a gMPNN$^{\bullet\bullet}$ can be bounded. Using the same overall framing as Maskey et al. [45, Lemma B.6],

**Corollary 7.** *Let $(\chi, d, \mu)$ be a metric-measure space and $W$ be a graphon s.t. Assumptions 1.1-3. are satisfied. Let $\Phi : \mathbb{R}^{2F} \to \mathbb{R}^H$ and $\Psi : \mathbb{R}^{F+H} \to \mathbb{R}^{F'}$ be Lipschitz continuous with Lipschitz constants $L_\Phi(M_{tr})$ and $L_\Psi(M_{tr})$. Consider a metric-space signal $f^{\bullet\bullet} : \chi \times \chi \to \mathbb{R}^F$ with $\|f^{\bullet\bullet}\|_\infty < \infty$. Let $p \in (0, \frac{1}{2(H+1)})$, and let $N \in \mathbb{N}$ satisfy (21). Suppose that $X_1, \ldots, X_N$ are drawn i.i.d. from $\mu$ in $\chi$, and then edges $A_{i,j} \sim Ber(W(X_i, X_j))$ i.i.d sampled. Then, condition (22) together with (24) below are satisfied in probability at least $1 - 2(H + 1)p$,*

$$\max_{i,j=1,\ldots,N} \|\Psi\Big(f^{\bullet\bullet}(\cdot, \cdot), M_A^{\bullet\bullet}\big(\Phi(f^{\bullet\bullet}, f^{\bullet\bullet})\big)(X_i, X_j)\Big) - \Psi\Big(f^{\bullet\bullet}(\cdot, \cdot), M_W^{\bullet\bullet}\big(\Phi(f^{\bullet\bullet}, f^{\bullet\bullet})\big)(X_i, X_j)\Big)\|_\infty$$

$$\leq L_\Psi(M_{tr})\Big(4\frac{\sqrt{2}\sqrt{\log(2N^2/p)}}{\sqrt{N}d_{cmin}^2}(L_\Phi(M_{tr})\|f^{\bullet\bullet}\|_\infty + \|\Phi(0,0)\|_\infty)$$

$$+ \frac{2\sqrt{2}(L_\Phi(M_{tr})\|f^{\bullet\bullet}\|_\infty + \|\Phi(0,0)\|_\infty)\sqrt{\log(2N^2/p)}}{d_{cmin}\sqrt{N}}\Big), \tag{24}$$

*Proof.* The proof is the same as Maskey et al. [45, Lemma B.6]. The difference comes from the different bound in our Lemma 9 and the bound used by Maskey et al. [45, Lemma B.5]. □

### G.2 Proof for Theorem 2

Finally, we can prove Theorem 2. The proof closely follows that of Maskey et al. [45, Theorem B.14], adapted to our setting. Using the same overall framing as Maskey et al. [45, Theorem B.14].

**Theorem 2** (OOD convergence without in-distribution convergence)**.** *For a random graph model $(W, f)$ satisfying Definition 1, let $N^{tr}$ be a random variable defining the distribution of graph sizes in training. Define the test distribution $(G^{te}, \boldsymbol{F}^{te}) \sim (W, f)$ through the causal graph in Figure 1 as an interventional change to obtain larger test graph sizes where $\min(supp(N^{te})) \gg M_{tr} = \max(supp(N^{tr}))$ (which means any test graph is much larger than the largest possible training graph). Let $\Theta = ((\Phi^{(l)})_{l=1}^T, (\Psi^{(l)})_{l=1}^T)$ be a MPNN as in Definition 2 with $T$ layers such that $\Phi^{(l)}$ and $\Psi^{(l)}$ that are learned from the training data and are Lipschitz continuous with Lipschitz constants $L_\Phi^{(l)}(M_{tr})$ and $L_\Psi^{(l)}(M_{tr})$. Let gMPNN$^{\bullet\bullet}$ $\Theta_W^{\bullet\bullet(T)}$ and cMPNN$^{\bullet\bullet}$ $\Theta_W^{\bullet\bullet(T)}$ be as in Definitions 9 and 10. For a random graph model $(W, f)$ as in Definition 1 with $d_{cmin} > 0$. Let $X_1^{te}, ..., X_{N^{te}}^{te}$ and $\boldsymbol{A}^{te}$ be as in Definition 1. Let $p \in (0, \frac{1}{\sum_{l=1}^T 2(H_l+1)})$. Then, if $\frac{\sqrt{N^{te}}}{\sqrt{\log(2(N^{te})^2/p)}} \geq \frac{4\sqrt{2}}{d_{cmin}}$, we have with probability at least $1 - \sum_{l=1}^T 2(H_l + 1)p$,*

$$\delta_{A\text{-}W}^{\bullet\bullet} = \max_{i,j=1,\ldots,N^{te}} \|\Theta_A^{\bullet\bullet(T)}(\boldsymbol{F}^{\bullet\bullet})_{i,j} - \Theta_W^{\bullet\bullet(T)}(f^{\bullet\bullet})(X_i^{te}, X_j^{te})\|_\infty \leq (C_3 + C_4\|f^{\bullet\bullet}\|_\infty)\frac{\sqrt{\log(2(N^{te})^2/p)}}{\sqrt{N^{te}}},$$

*where the constants $C_3$ and $C_4$ are defined in the Appendix and depend on $\{L_\Phi^{(l)}(M_{tr}), L_\Psi^{(l)}(M_{tr})\}_{l=1}^T$ and the distribution of $(G^{tr}, \boldsymbol{F}^{tr})$.*

*Proof.* In this case, $\|\Phi^{(l)}(0,0)\|_\infty, \|\Psi^{(l)}(0,0)\|_\infty$ can be determined by $(G^{tr}, \boldsymbol{F}^{tr}), N^{tr}$ if the MPNN $\Theta$ has been trained on the training graph $(G^{tr}, \boldsymbol{F}^{tr})$.

Following the procedure of Maskey et al. [45, Appendix B.2] with Corollary 7, we can derive similarly, with probability at least $1 - \sum_{l=1}^T (2H_l + 1)p$,

$$\delta_{A\text{-}W}^{\bullet\bullet} \leq \sum_{l=1}^T L_\Psi^{(l)}(M_{tr})\Big(4\frac{\sqrt{2}\sqrt{\log(2(N^{te})^2/p)}}{\sqrt{N^{te}}d_{cmin}^2}(L_\Phi^{(l)}(M_{tr})\|f^{\bullet\bullet(l)}\|_\infty + \|\Phi^{(l)}(0,0)\|_\infty)$$

$$+ \frac{2\sqrt{2}(L_\Phi^{(l)(M_{tr})}\|f^{\bullet\bullet(l)}\|_\infty + \|\Phi^{(l)}(0,0)\|_\infty)\sqrt{\log(2(N^{te})^2/p)}}{d_{cmin}\sqrt{N^{te}}}\Big) \tag{25}$$

$$\prod_{l'=l+1}^T ((L_\Psi^{(l')}(M_{tr}))^2 + \frac{8}{d_{cmin}^2}(L_\Phi^{(l')}(M_{tr}))^2(L_\Psi^{(l')}(M_{tr}))^2),$$

Using the same proof in Maskey et al. [45, Lemma B.9], we can derive

$$||f^{\bullet\bullet(l)}||_\infty \leq B_1^{\bullet\bullet(l)} + B_2^{\bullet\bullet(l)}||f||_\infty,$$

where $B_1{}^{\bullet\bullet(l)}$, $B_2{}^{\bullet\bullet(l)}$ are independent of $f^{\bullet\bullet}$, and

$$B_1^{\bullet\bullet(l+1)} = \sum_{k=1}^{l+1} \left(L_\Psi^{(k)}(M_{\mathrm{tr}})\frac{1}{d_{\mathrm{cmin}}}\|\Phi^{(k)}(0,0)\|_\infty + \|\Psi^{(k)}(0,0)\|_\infty\right)$$
$$\prod_{l'=k+1}^{l+1} L_\Psi^{(l')}(M_{\mathrm{tr}})\left(1 + \frac{1}{d_{\mathrm{cmin}}}L_\Phi^{(l')}(M_{\mathrm{tr}})\right) \tag{26}$$

and

$$B_2^{\bullet\bullet(l+1)} = \prod_{k=1}^{l+1} L_\Psi^{(k)}(M_{\mathrm{tr}})\left(1 + \frac{1}{d_{\mathrm{cmin}}}L_\Phi^{(k)}(M_{\mathrm{tr}})\right). \tag{27}$$

Now we can decompose the summation in Equation (25). First, we defince $C_3$ as

$$C_3 = \sum_{l=1}^{T} L_\Psi^{(l)}(M_{\mathrm{tr}})\left(4\frac{\sqrt{2}}{d_{\mathrm{cmin}}^2}(L_\Phi^{(l)}(M_{\mathrm{tr}})B_1^{\bullet\bullet(l)} + \|\Phi^{(l)}(0,0)\|_\infty)\right.$$
$$\left. + \frac{2\sqrt{2}(L_\Phi^{(l)}(M_{\mathrm{tr}})B_1^{\bullet\bullet(l)} + \|\Phi^{(l)}(0,0)\|_\infty)}{d_{\mathrm{cmin}}}\right) \tag{28}$$
$$\prod_{l'=l+1}^{T} ((L_\Psi^{(l')}(M_{\mathrm{tr}}))^2 + \frac{8}{d_{\mathrm{cmin}}^2}(L_\Phi^{(l')}(M_{\mathrm{tr}}))^2(L_\Psi^{(l')}(M_{\mathrm{tr}}))^2),$$

Then we can define $C_4$ as

$$C_4 = \sum_{l=1}^{T} L_\Psi^{(l)}(M_{\mathrm{tr}})\left(4\frac{\sqrt{2}}{d_{\mathrm{cmin}}^2}L_\Phi^{(l)}(M_{\mathrm{tr}})B_2^{\bullet\bullet(l)} + \frac{2\sqrt{2}L_\Phi^{(l)}(M_{\mathrm{tr}})B_2^{\bullet\bullet(l)}}{d_{\mathrm{cmin}}}\right)$$
$$\prod_{l'=l+1}^{T} ((L_\Psi^{(l')}(M_{\mathrm{tr}}))^2 + \frac{8}{d_{\mathrm{cmin}}^2}(L_\Phi^{(l')}(M_{\mathrm{tr}}))^2(L_\Psi^{(l')}(M_{\mathrm{tr}}))^2), \tag{29}$$

It is clear to see we can rewrite Equation (25) as

$$\delta_{\mathrm{A\text{-}W}}^{\bullet\bullet} \leq (C_3 + C_4\|f^{\bullet\bullet}\|_\infty)\frac{\sqrt{\log(2(N^{\mathrm{te}})^2/p)}}{\sqrt{N^{\mathrm{te}}}}. \tag{30}$$

Thus $C_3$ and $C_4$ depends on $\{L_\Phi^{(l)}(M_{\mathrm{tr}})\}_{l=1}^{T}$ and $\{L_\Psi^{(l)}(M_{\mathrm{tr}})\}_{l=1}^{T}$ and possibly on $(G^{\mathrm{tr}}, \boldsymbol{F}^{\mathrm{tr}})$ and $N^{\mathrm{tr}}$. □