# OpenReview forum: "OOD Link Prediction Generalization Capabilities of Message-Passing GNNs in Larger Test Graphs"
_NeurIPS.cc/2022/Conference — NeurIPS 2022 Accept_

### Official Review · Reviewer_FCSF · 2022-06-26

**Rating:** 7
**Confidence:** 4
**Soundness:** 3 good
**Presentation:** 3 good
**Contribution:** 3 good

**Summary:**

This paper studies the out-of-distribution (OOD) inductive link prediction capability of graph message passing neural networks (gMPNNs) in terms of graph sizes. While showing gMPNNs predictions can converge to random guesses when test graphs get larger, they also propose to use pairwise embeddings which can provably resolve the issue. The theoretical results are validated with synthetic experiments on random graphs.

**Questions:**

The authors conduct extensive works on the theoretical analysis which is appreciated. However, the theoretical tools used in the paper are similar to [1], and the solution is not new in the literature [2]. There are several concerns raised regarding the scope and novelty of both the theoretical and empirical results:

The scope studied in this work seems to be limited:
- *Task*: This paper studies the OOD inductive link prediction task in terms of graph sizes. However, it seems many of existing link prediction applications, such as knowledge graph and recommender system, focus more on the transductive settings [3,4], or ``inductive’’ settings where the training graph is a subgraph of the test graph, which is different from the setting studied in the paper. It would further strengthen the paper if the authors could provide more concrete examples for the inductive applications studied in the paper, or analysis on other settings.

- *Distribution shifts*: Moreover, the distribution shifts in graphs can be more complicated than graph sizes, as discussed in [5,6,7,8]. It would make the paper more impactful if the authors could also discussed other potential distribution shifts, such as covariate shifts on node features, subgraph appearance frequencies, averaged graph degrees [8].

- *Graph model*: The main results seem to rely largely on a certain graph model, i.e., SBM. And the failure cases seem to be limited to certain nodes in the model. I understood that providing general results for all graph models would be too difficult, however, if authors could provide more results for other graph models or more realistic graphs, it would further strengthen the paper by providing more guide for practitioners as well as future works built on the results of this work.


Regarding the novelty:
- The hardness example seems to rely on the convergence results of gMPNNs to cMPNNs in terms of the node embeddings. Does the similar results can also be developed for graph classification and node classification, too?

- Since the main results of this paper largely rely on the convergence of gMPNNs to cMPNNs on node embeddings, which seems to be similar to Theorem 3.1 from [1]. More discussions on the differences seem to be expected, especially when the convergence can also be applied to other tasks. Otherwise the results for link prediction seem to be a straightforward extension of [1] which may weaken the novelty of the work.

- To mitigate the issue of gMPNNs when generalizing to larger test graphs, the authors propose to use pairwise embeddings [2]. Does it mean using the original pairwise node embeddings [2] can already solve the issue? It would be more helpful for the future works built on the paper if the authors could summarize the implications of the theoretical results to the practitioners.

Minor points:
- The causal graph seems not to provide further implications.
- In line 190, i \in S_A should be I \in S_a .

== post rebuttal ==
The authors adequately resolve my concerns during the rebuttal. Hence I'd like to raise my ratings to 7.

References:

[1] Sohir Maskey, Yunseok Lee, Ron Levie, and Gitta Kutyniok. Stability and generalization capabilities of message passing graph neural networks. arXiv 2022.

[2] Balasubramaniam Srinivasan and Bruno Ribeiro. On the equivalence between positional node embeddings and structural graph representations. ICLR 2020.

[3] Shaoxiong Ji, Shirui Pan, Erik Cambria, Pekka Marttinen and Philip S. Yu. A Survey on Knowledge Graphs: Representation, Acquisition, and Applications. TNNLS 2022.

[4] Víctor Martínez, Fernando Berzal, and Juan-Carlos Cubero. A Survey of Link Prediction in Complex Networks. ACM Computing Survey 2017.

[5] Beatrice Bevilacqua, Yangze Zhou, and Bruno Ribeiro. Size-invariant graph representations for graph classiﬁcation extrapolations. ICML 2021.

[6] Yingxin Wu, Xiang Wang, An Zhang, Xiangnan He, Tat-Seng Chua. Discovering Invariant Rationales for Graph Neural Networks. ICLR 2022.

[7] Qitian Wu, Hengrui Zhang, Junchi Yan, David Wipf. Handling Distribution Shifts on Graphs: An Invariance Perspective. ICLR 2022.

[8] Yongqiang Chen, Yonggang Zhang, Yatao Bian, Han Yang, Kaili Ma, Binghui Xie, Tongliang Liu, Bo Han, James Cheng. Invariance Principle Meets Out-of-Distribution Generalization on Graphs. arXiv 2022.


**Limitations:**

Since this is a theoretical work, potential negative social impacts may not be applicable.

**Strengths And Weaknesses:**

*Originality & Significance*: Studying the OOD ability of gMPNNs is of great importance to the community, though the theoretical tools used in the paper are similar to Maskey et. al., 2022, and the solution is not new in the literature (Srinivasan and Ribeiro, 2020).

*Quality*: The authors conduct extensive works on the theoretical analysis which is appreciated. However, there are several concerns raised regarding the scope and novelty of both the theoretical and empirical results. See the Questions below for more details.

*Clarity*: The paper is well-written and organized. The authors provide informative explanations for each theoretical result which makes the paper easy to follow despite lots of theories.

---

> ### Author Response · Authors · 2022-08-02
> **Rebuttal**
>
> We thank the reviewer for the comments and questions. These will help improve the paper and allow us to discuss important aspects of the paper.
>
> *Q1. What are the differences to Maskey et. al., 2022?*
>
> The key differences are:
>
> Added a different type of result: We have two analyses: One for degree normalized aggregations (as in Maskey et. al.,) and another one for GIN-type GNNs which are unnormalized aggregations (different proofs and probably more expressive as the GIN paper shows). We prefer the results with aggregations that are not degree-normalized (which do not need the proof results of Maskey et. al.,) since these are more expressive. These are unrelated to the results of Theorem 3.1 of Maskey et. al..
>
> Extension: Maskey et. al., 2022 also considers mean-field edges (i.e., not sampled) while we consider sampled edges. Adding sampling makes the analysis a little more complex (which resulted in a longer appendix).
>
> Another key aspect: This is described in a footnote in the new draft. Maskey et. al., misunderstood the implications of its Theorem 3.1 as an in-distribution convergence result. We correctly show that this is about OOD convergence. Once the GNN learns a given function for a given set of graph sizes in training, the proof shows how that learned gMPNN* will converge to the cMPNN output once the graph becomes larger in test. The authors of Maskey et. al. are now aware of this key difference. Hence, Maskey et. al. had to change most proofs due to this insight, which completely changed the paper, now titled “Generalization Analysis of Message Passing Neural Networks on Large Random Graphs” (https://arxiv.org/pdf/2202.00645v5.pdf) from “Stability and Generalization Capabilities of Message Passing Graph Neural Networks” (2202.00645v3.pdf, which is the version we used at the time). Very little of the math we cited has survived in v5 of Maskey et. al.. Since arXiv is permanent, we changed our citation to arXiv:2202.00645v3 rather than just using arXiv:2202.00645 and added a footnote to describe the reason for the change. Overall, this goes to show that the insights in our paper are not that obvious.
>
> *Q2: The solution is not new in the literature, see (Srinivasan and Ribeiro, 2020)*
>
> (Srinivasan and Ribeiro, 2020) provides an existence proof that joint representations exist and can perform link prediction. Srinivasan and Ribeiro never described how to do them. We provide a constructive & counterintuitive joint representation. Why is our joint representation counterintuitive? Until now, all joint representations of a pair (u,v) would include (u,i) representations of the previous layer if (u,i) had an edge, $u,v,i \in V$. Our representation does not work like that. Consider Alice, Bob, and Carol in a social network. In the representation of (Alice,Bob) we add the previous layer representation of (Alice,Carol) if Bob is friends with Carol) (not if Alice is friends with Carol). This is driven by the theory and defied our intuition. This is what we needed to show that the representation is counterfactually-invariant in Theorem 2. This novel representation works OOD in our SBM tasks and in a new real-world task (for the rebuttal) using ogbl-ddi dataset (Table 3 of revised draft). We used ogbl-ddi since this was the only graph dataset we could run all baselines in time for the rebuttal.
>
> *Q3a: The distribution shifts in graphs can be more complicated than graph sizes, as discussed in [5,6,7,8].*
>
> *Q3b: The causal graph seems not to provide further implications*
>
> Theorem 2 bounds is about showing that our gMPNN** is an approximate counterfactual-invariant representation according to our causal DAG. Without the DAG we would not be able to show that gMPNN** is robust OOD. **We will expand this causal language in the camera-ready, once we have an extra content page.** We also included a short discussion of [5,6,7,8] in the appendix of the new draft. We point out that these works differ in that they focus on node classification and graph classification, while link prediction tasks are significantly different (Shrinivasan and Ribeiro, 2020 explains this difference in more details).
>
> *Q4: It would further strengthen the paper if the authors could provide more concrete examples for the inductive applications studied in the paper, or analysis on other settings.*
>
> Most industrial applications of link prediction want to train the predictor on an induced subgraph (to save computational costs) and then apply the predictor on the whole graph. For the rebuttal we performed a similar experiment on a real-world graph (ogbl-ddi), where training uses an induced subgraph with only 10% of the nodes. The results match that of our theory and our SBM results (see Table 3 in the appendix). gMPNN** has invariant OOD performance, while permutation-equivariant GNNs suffer significant loss in performance. We also added experiments from large training graphs to small test graphs (Table 4 in the Appendix).

---

> > ### Author Response · Authors · 2022-08-06
> > **Window to update draft running out**
> >
> > Dear Reviewer,
> >
> > The window to update our draft running out. Please let us know ASAP if you feel other changes are needed to address your concerns.
> >
> > Thanks,
> > The Authors

---

> > > ### Author Response · Authors · 2022-08-06
> > > **Appendix update**
> > >
> > > Dear Reviewer,
> > >
> > > Please note that some updates (including all Tables) are in the Appendix due to the rebuttal page limit constraints. In Camera Ready will move Table 1 to the main paper together with some of the changes now in the Appendix.
> > >
> > > Thank you for your time,
> > > Authors

---

> > ### Comment · Reviewer_FCSF · 2022-08-06
> > **Reply to authors**
> >
> > Dear authors,
> >
> > Thanks for your detailed reply. The newly added discussions of related works and the experiments resolve all of my concerns. I'll reconsider my ratings.

---

### Official Review · Reviewer_BWxA · 2022-07-10

**Rating:** 7
**Confidence:** 3
**Soundness:** 3 good
**Presentation:** 3 good
**Contribution:** 3 good

**Summary:**

This work studies the ability of message passing neural network (MPNN) to perform prediction tasks on a test graph larger than the train graph. It first gives a definition of random graph and continuous MPNN (cMPNN) and proves that as the test random graph size approaches infinity, gMPNN converges to cMPNN. Then authors focus on Stochastic Block Model (SBM) and illustrate how the convergence leads to bad generalization on SBM: node representations of distinct SBM blocks can become increasingly similar as the test graph size grows even if they are non-isomorphic with a high probability. Then authors introduce structural pairwise embedding and prove that this method can solve the problem. Empirical results on SBM verify the theoretical analysis.

**Questions:**

Is there any results for random graphs other than SBM?

**Limitations:**

Not related.

**Strengths And Weaknesses:**

Strength:

1. Good originality and significance: the first theoretical study on the ability of GNNs to perform inductive out-of-distribution (OOD) link prediction tasks.

2. Novel theoretical analysis.

3. Clear representation.

Weakness:

The theoretical analysis and experiments are constrained to SBM. However, real-world graphs are different. For example, some graphs have no node features.

---

> ### Author Response · Authors · 2022-08-02
> **Rebuttal**
>
> We thank the reviewer for the support and good feedback.
>
> *Q. Are there any results for random graphs other than SBM? For example, some graphs have no node features.*
>
> In the synthetic experiments, we used the empirical size-normalized degrees (lines 659-660 in the updated draft) as node features.  During the rebuttal, we have performed new experiments on the real-world graph ogbl-ddi (https://ogb.stanford.edu/docs/linkprop/#ogbl-ddi), which is a homogeneous, unweighted, undirected graph, representing the drug-drug interaction network with no node features. Similar to our synthetic experiments, we also use the empirical size-normalized degrees as node features.
>
> We choose this graph also because we assume the underlying drug-drug interaction might approximately follow an SBM model. For training, we only select 10% of the nodes and its induced subgraph.  As shown in Table 3 in the new draft, most of the MPNN GNN baselines have decent transductive performance but poor OOD performance, where our proposed joint representation gMPNN** achieves good results in in-distribution and OOD inductive tasks, matching our theory. Interestingly, in a real-world graph we see that a learned gMPNN** performs significantly better than the fixed gMPNN** (as one would expect). Also, since the OOD graph is larger, the gMPNN** predictor is able to achieve higher confidence OOD (also expected on harder tasks). As shown in Table 3, most of the MPNN GNN baselines still suffer in the inductive performance (both in-distribution and OOD), where our proposed gMPNN** achieves consistent results in in-distribution and OOD inductive tasks, showing that our insights are not limited to SBM models.
>
> In addition to the questions, we restated our Theorem 2 to add language explaining that the bound shows gMPNN** is an approximate counterfactually-invariant representation. We will add more causal language to help readers understand how our results are connected to OOD tasks in the camera-ready, where we will have an extra page available.

---

### Official Review · Reviewer_Bs4U · 2022-07-11

**Rating:** 4
**Confidence:** 3
**Soundness:** 2 fair
**Presentation:** 4 excellent
**Contribution:** 3 good

**Summary:**

This paper theoretically analyze the GNN's capalability for handling link prediction. The main results show that using node embeddings for link prediction could fail to distinguish two nodes even though they are distinct in the graph as node numbers increase, and using pairwise (edge) embeddings for link prediction could successfully learn a correct prediction model and generalize the OOD graphs with larger sizes. Experiments on synthetic datasets verify the theoretical results.

**Questions:**

While the main body of the paper seems interesting, there are still some confusing parts which would impact how the contributions are interpreted and need to be clarified before a proper evaluation could be made. I hope the following the questions could be properly addressed during the rebuttal.

1. As far as I understand, the key result of Section 3.2 is Prop 1 (though it is claimed to be contained in the Appendix) which cliams to show that for two nodes which are even distinct in the graph the link prediction model would also map them into the same embedding as the node numbers increase. If this holds true, such a result implies that the models based on node embeddings may fail to generalize to larger graphs. However, I fail to find Prop 1 in the Appendix, so cannot check its soundness and could overally downweigh the significance/contribution of Section 3.2.

2. In Section 4, the results show that the models based on pairwise embeddings would converge to an "ideal" model whose stationary solution is the graphon as node numbers increase. Then a conclusion is made that the models would successfully generalize to larger graphs. I am a little confused at this point: 1) what is the implication of the stationary point, does it mean one of the optimums or the unique one? 2) the stationary point is obtained from assumptions for the specific forms of \Phi and \Psi, does it mean the conclusion only holds for very specific situation and cannot be applied to the general regime? 3) why such a stationary point implies desirable OOD performance, it seems these two have no clear connection.

**Ethics Review Area:**

["I don’t know"]

**Limitations:**

The major limitation lies in the experiments, though most of contents seem reasonable and sound. The experiment datasets only cover synthetic datasets which are simple and limited. I suggest adding more experiments on real-world datasets, especially some common GNN benchmarks for link prediction and comparing with some SOTA approaches (even though they are not designed for OOD regime). Though I understand that the main focus of this paper lies in the theoretical insights and perspecitve, more results on benchmarks and comparison with more strong models (currently used baselines seem too weak and simple) could make this work more complete and convincing, especially that one of the contributions of this work is a new approach for OOD link prediction.

Also, there are some recent works on out-of-distribution generalization on graphs, e.g., [1, 2, 3], which are missing in the related works and need to be discussed.

[1] Size-Invariant Graph Representations for Graph Classification Extrapolations, ICML 21.

[2] Handling Distribution Shifts on Graphs: An Invariance Perspective, ICLR 22.

[3] From Local Structures to Size Generalization in Graph Neural Networks, ICML 21.

**Strengths And Weaknesses:**

Strengths: 1. The insight of this paper is helpful for understanding the capablity and behaviors of GNN models for OOD generalization link prediction tasks, which is an active and open problem in the community. 2. Most of the results are intuitive and see correct though I did not carefully check the proof. 3. The paper is well written and easy to follow with very clear and clean notations.

Weaknesses: 1. Some of the results are confusing and need more illustration (see comments below). 2. Experiments are only conducted on synthetic datasets and more results on real-world datasets especially common GNN benchmarks are expected.

---

> ### Author Response · Authors · 2022-08-02
> **Rebuttal**
>
> We thank the reviewer for the great questions and good feedback, and for enjoying reading our paper. We address the key questions below with the changes we made to our draft.
>
> *Q1. The key results seem to be Prop 1... But failed to find it in the appendix.*
>
> It is in Appendix D (page 21 of original submission and 22 of new draft). Proposition 1 shows that even isomorphic SBMs are likely asymmetric (page 22 of original and 24 of new draft).
>
> Q2. How does the stationary point result in pair-wise embeddings lead to the conclusion that it can be generalized to OOD regimes?
>
> In a link prediction task (Definition 8), given the input as a pair of nodes (x,y), in the context of graphon random graph model (Definition 1), the goal is to learn a function f(x,y) that approximates W(x,y), since edges between (x,y) is sampled with probability W(x,y) (Definition 1). In our new draft we restated our Theorem 2 to add language explaining that the bound shows gMPNN** is an approximate counterfactually-invariant representation. We will add more causal language to help readers understand how our results are connected to OOD tasks. We also included a more in-depth discussion of the causal DAG with respect to other possible covariate shifts on graphs. See Mouli and Ribeiro, 2021 [33] and Veitch et al., 2021 [46] for a definition of counterfactually-invariant classifier (which will be added in the CR's extra page).
>
> *Q2.1 What is the implication of the stationary point, does it mean one of the optimums or the unique one?*
>
> Given the structure of the cMPNN**, there exists functions of \Phi and \Psi (infinity choices as stated in the proof of Lemma 2 in page 44 of the new draft), that the output of cMPNN** converge to the graphon functions W(x,y). In Theorem 2, we have shown that gMPNN** is an approximate counterfactually-invariant representation since converge to cMPNN** when the size of the test graph increases. And since the optimization goal for the loss function is to predict W(x,y), this results implies that the gMPNN** can approximately output the graphon functions W(x,y) with some \Phi and \Psi as one of the optima (but not the unique one). Since the optimization is non-convex, we include experiments to showcase its ability to achieve an approximate counterfactually-invariant predictor (approximately invariant under changes in graph size).
>
> *Q2.2. The stationary point is obtained from assumptions for the specific forms of \Phi and \Psi, does it mean the conclusion only holds for very specific situations and cannot be applied to the general regime?*
>
> There are many choices of \Psi and \Phi that satisfy our conditions. The conclusion also holds for a more general regime illustrated in our experimental section. In all our experiments, we include both settings where \Psi and \Phi are fixed functions in Lemma 2, and \Psi and \Phi are a learnable feedforward neural network (lines 355-356 in the updated version). Empirically, the learnable \Psi and \Phi gives the best performance in all experiments, including the new experiments conducted for real-world graphs (detailed later in further responses) in the updated draft.
>
> *Q 2.3. Why does such a stationary point imply desirable OOD performance?*
>
> In our causal graph W(x,y) is invariant to graph size. OOD size invariance requires causal assumptions, see Bevilacqua et al. [5].
>
> *Q3. Suggest to add more experiments on real-world datasets, especially some common GNN benchmarks for link prediction and comparing with some SOTA approaches (even though they are not designed for OOD regime)*
>
> For the rebuttal we performed experiments on a real-world graph (ogbl-ddi), which represents a drug-drug interaction network. We choose this graph due to its size and the short rebuttal timeline. For training, we only select 10% of the nodes and its induced subgraph.  As shown in Table 3 in the new draft, most of the MPNN GNN baselines have decent transductive performance but poor OOD performance. Our proposed gMPNN** achieves good results in OOD tasks, matching our theory in a real-world dataset. In a real-world graph we see that a learned gMPNN** performs significantly better than the fixed gMPNN** (as one would expect). Also, since the OOD graph is larger, the gMPNN** predictor is able to achieve higher confidence OOD (also expected on harder tasks). The conclusions match that of the SBM model. We did not have enough time to add other link prediction baselines (these are expected to fail, hence they were not high-priority for the rebuttal but will be added to the camera-ready).
>
> *Q4. There are some recent works on out-of-distribution generalization on graphs, e.g., [1, 2, 3], which are missing in the related works and need to be discussed*
>
> Thank you for pointing out these works. We had discussed [1] and [3] in the introduction (line 79-80 in the original draft) and further related work (line 577-578 in the original draft). We added [2] to related work in the new draft (line 605-610).

---

> > ### Author Response · Authors · 2022-08-06
> > **Window to update draft running out**
> >
> > Dear Reviewer,
> >
> > The window to update our draft running out. Please let us know ASAP if you feel other changes are needed to address your concerns.
> >
> > Thanks,
> > The Authors

---

> > > ### Author Response · Authors · 2022-08-06
> > > **Appendix update**
> > >
> > > Dear Reviewer,
> > >
> > > Please note that some updates (including all Tables) are in the Appendix due to the rebuttal page limit constraints. In Camera Ready will move Table 1 to the main paper together with some of the changes now in the Appendix.
> > >
> > > Thank you for your time,
> > > Authors

---

### Official Review · Reviewer_3cpo · 2022-07-13

**Rating:** 5
**Confidence:** 4
**Soundness:** 3 good
**Presentation:** 3 good
**Contribution:** 3 good

**Summary:**

In this paper, the authors aims to analyze the limitations of existing MPNNs on performing link prediction. Specifically, they identify that existing node-based MPNNs are not able to handle out-of-distribution (OOD) graphs for inductive link prediction, where the ``distribution'' is related to node size. Then, they further propose an edge-based MPNN to overcome this issue.

**Questions:**

1. It is not very clear whether it is really an OOD issue. It seems that node-based MPNNs will experience difficulty to learn distinguishable node representations for nodes from distinct isomorphic blocks when the size of the test graph increases, regardless of the training graph. So, is it an OOD issue or an issue of learning distinguishable representations for isomorphic nodes? It would be better if the authors could provide more detailed explanations on this. For example, is there an OOD issue when the test graph is smaller than the training graph? If the training graph and test graph are both large, will the test performance be good?

2. A large portion of the analyses are based on a specific kind of SBM model, where isomorphic blocks exist. It seems the OOD issue identified in this paper is strongly related to isomorphic blocks. So, does the OOD issue exist in other kinds of graphs (for example, graphs generated from SBM model without isomorphic blocks)?
Also, there are some other papers[1] suggesting that MPNNs are not able to deal with graphs with isomorphic patterns.  It would be better if the authors could provide a discussion and comparison with these existing works.

3. In general, it would be better if the authors could provide a more detailed and accurate description on the OOD issue that they try to address. As mentioned in the first two points, the analyses seems to be limited to very specific cases while the title and the introduction is very general.


[1] Labeling Trick: A Theory of Using Graph Neural Networks for Multi-Node Representation Learning

**Strengths And Weaknesses:**

Strengths
1. The authors conduct theoretical analysis to study the OOD issue for inductive link prediction when the test graph is large.
2. The authors conducted comprehensive results to verify the correctness of their theories.
3. A new framework is proposed to address the OOD issue for the link prediction task.

Weakness
1. A large portion of the analyses are based on the SBM model. Also, the features for nodes in the same block are assumed to be the same (Definition 6). Furthermore, the analyses are generally based on SBM models with isomorphic blocks. It is not clear how these analyses can be extended to more general cases.
2. It is not very clear whether it is really an OOD issue. It seems that node-based MPNNs will experience difficulty to learn distinguishable node representations for nodes from distinct isomorphic blocks when the size of the test graph increases, regardless of the training graph.

---

> ### Author Response · Authors · 2022-08-02
> **Rebuttal**
>
> We thank the reviewer 3cpo for the great questions and good feedback. We note that our paper significantly deepens our understanding of OOD link prediction.
>
> *Q1a. Is this an OOD issue or an issue of learning distinguishable representations for isomorphic nodes?*
> *Q1b. Why most results rely on SBMs with isomorphic blocks?*
> *Q1c. It would be better if the authors could provide a more detailed and accurate description on the OOD issue that they try to address*
>
> Great questions. We added a more in-depth discussion to the revised draft. The isomorphic SBM offers a way to illustrate an OOD issue in link prediction with equivariant GNNs, and design new counterfactually-invariant link prediction methods. In isomorphic graphs, and for isomorphic node pairs, both You et al. and Srinivasan and Ribeiro have shown that equivariant GNN link prediction can fail (which “Labeling Trick” and other recent work have used as insights for their approaches. “Labeling Trick”’s Theorem 2 relies on isomorphic nodes (see their proof)). But isomorphic nodes are rare in practice (see Figure 3 in the Appendix) and in large random graphs.
>
> An important open question is then if equivariant GNN would be able to predict links in asymmetric graphs (with non-isomorphic nodes). That is, the concerns of You et al., Srinivasan and Ribeiro, and “Labeling trick” may not be important in practice. We devised an interesting task to test this: The isomorphic SBM is symmetric (in average) but the resulting sampled random graph is likely asymmetric  (Proposition 1). Our results show that the concerns of You et al. and Srinivasan and Ribeiro (and “Labeling trick”) are not really a big deal for in-distribution link prediction tasks.
>
> However, under the same setting, the OOD can be an issue even in asymmetric graphs (with a symmetric underlying model). This deepens our understanding of the issues in link prediction (graph asymmetries are no longer enough to allow the use of equivariant GNNs if the task is OOD). We restated our Theorem 2 to add language explaining that the bound shows gMPNN** is an approximate invariance. For the camera-ready (when we have 10 pages) we will expand the connection between the causal model and the OOD task (we had added it for the rebuttal but had to undo it since it made the paper go over the 9-page limit of the rebuttal).
>
> Beyond the theory, our revised draft performs new OOD link prediction experiments. One of them uses real data (Table 3 in the revised draft, using ogbl-ddi which is our smallest real-world graph that we had time to run all our experiments for the rebuttal). The results show that the isomorphic SBM model showcases a phenomenon that happens in real-world data.
>
>
> *Q2: It seems that node-based MPNNs will experience difficulty to learn distinguishable node representations for nodes from distinct isomorphic blocks when the size of the test graph increases, regardless of the training graph.*
>
> Not quite, we will make sure this is explained in more detail in the camera-ready. In training, the GNN learns to make graph asymmetries salient-enough to perform link prediction (see in-distribution results in Table 1). Figure 2 shows that between 2k and 20k nodes a graph is large enough to get very similar representations if the GNN weights were randomly initialized. That is, the ability to predict links in Table 1 rests in the training (we have analyzed this numerically but can include a plot in the paper if the reviewer thinks it will help clarify the message). However, we show that the salient asymmetries the GNN found for a given graph size in training become non-salient as the test graph grows. This is the reason for the nearly-random performance of equivariant GNNs in the OOD task in Table 1.
>
> In the updated draft, we also perform training on larger graphs (training graph with 10^4 nodes), and the in-distribution inductive performance on 10^4 sized graph (Table 4 in the revised draft) is much better than the OOD inductive performance on 10^4 sized graph when the training graph has only 10^3 nodes (Table 1), empirically showing the training graph matters and that the OOD link prediction challenge also happens when test graphs shrink in size (surprisingly, GCN is the permutation-equivariant GNN method that least suffers from this OOD change. Equivariant GNNs are, however, no match to our gMPNN** method which is designed to be approximately invariant OOD.).
>
> *Q3: The features for nodes in the same block are assumed to be the same (Definition 6).*
>
> This assumption is just to make the analysis simpler. In our experiments we used the empirical size-normalized degrees, which are stochastic features (not constant) in the sampled graph.
>
> *Q4. Does the OOD issue exist in other kinds of graphs (for example, graphs generated from SBM model without isomorphic blocks)?*
>
> For the rebuttal we performed experiments on a real-world graph (ogbl-ddi) and we generally see the same behavior as in the SBM (but less pronounced).

---

> > ### Author Response · Authors · 2022-08-06
> > **Window to update draft running out**
> >
> > Dear Reviewer,
> >
> > The window to update our draft running out. Please let us know ASAP if you feel other changes are needed to address your concerns.
> >
> > Thanks,
> > The Authors

---

> > > ### Author Response · Authors · 2022-08-06
> > > **Appendix update**
> > >
> > > Dear Reviewer,
> > >
> > > Please note that some updates (including all Tables) are in the Appendix due to the rebuttal page limit constraints. In Camera Ready will move Table 1 to the main paper together with some of the changes now in the Appendix.
> > >
> > > Thank you for your time,
> > > Authors

---

> > > > ### Comment · Reviewer_3cpo · 2022-08-08
> > > > **Thank you for the reply**
> > > >
> > > > Dear authors,
> > > >
> > > > Thanks for your detailed reply. Some of the concerns are addressed. However, the major concern that "A large portion of the (theoretical) analyses are based on the SBM model with isomorphic blocks" is not addressed. Hence, I will adjust my rating to 5.

---

### Meta-Review · Area_Chair_JRbm · 2022-08-25

**Recommendation:** Accept
**Confidence:** Less certain

**Metareview:**

This paper makes a compelling contribution to the study of size generation in the inductive link prediction setting. The authors extend previous results on size generalization for graph classification in a non-trivial manner and propose a new MPNN architecture with improved theoretical properties. Though the analysis is based on contrived random graph models (graphons), it's very encouraging to see that the proposed changes to the MPNN architecture lead to better experimental results. The authors also made a good effort in the rebuttal to explain their work and to enhance their experiments following the reviewers' feedback.

**Award:**

No

---

### Decision · Program_Chairs · 2022-09-14

Accept